# Recent acceleration in global ocean heat accumulation by mode and intermediate waters

Zhi Li [1,2,3] ✉, Matthew H. England [2,3] & Sjoerd Groeskamp [4]

The ocean absorbs >90% of anthropogenic heat in the Earth system, moderating global atmospheric warming. However, it remains unclear how this heat uptake is distributed by basin and across water masses. Here we analyze historical and recent observations to show that ocean heat uptake has accelerated dramatically since the 1990s, nearly doubling during 2010–2020 relative to 1990–2000. Of the total ocean heat uptake over the Argo era 2005–2020, about 89% can be found in global mode and intermediate water layers, spanning both hemispheres and both subtropical and subpolar mode waters. Due to anthropogenic warming, there are significant changes in the volume of these water-mass layers as they warm and freshen. After factoring out volumetric changes, the combined warming of these layers accounts for ~76% of global ocean warming. We further decompose these water-mass layers into regional water masses over the subtropical Pacific and Atlantic Oceans and in the Southern Ocean. This shows that regional mode and intermediate waters are responsible for a disproportionate fraction of total heat uptake compared to their volume, with important implications for understanding ongoing ocean warming, sea-level rise, and climate impacts.

The ocean directly impacts the Earth's climate by absorbing and redistributing large amounts of heat, freshwater, and carbon, and by exchanging these properties with the atmosphere[1]. About 91% of the excess heat trapped by greenhouse gases[1] and 31% of human emissions of carbon dioxide[2] are stored in the ocean, shielding humans from even more rapid changes in climate. However, warmer oceans result in sea-level rise, ice-shelf melt, intensified storms, tropical cyclones, and marine heatwaves, as well as more severe marine species and ecosystem damage[1,3,4]. These effects depend on the pattern of ocean warming; it is thus critical to quantify the dynamics and distribution of ocean warming to better understand its consequences and predict its implications[5].

The observed distribution of ocean warming is not uniform. About 90% of total ocean warming is found in the upper 2000 m, with over two-thirds concentrated in the upper 700 m since the 1950s[5–8], and an increase of warming rates at both intermediate depths of 700–2000 m[5,7,9–12], and in the deeper ocean below 2000 m[5,7,10,11]. The Southern Ocean south of 30°S has been estimated to account for 35–43% of global ocean warming from 1970 to 2017[13], and an even greater proportion in recent years[12,13], while Northern Hemisphere ocean warming appears to be concentrated in the Atlantic Ocean[6,14,15]. Due to the accumulated excess heat in ocean basins, an acceleration of total ocean warming has become more evident from recent observational-based studies[5,6,16,17]. While much past work has focused on the distribution of ocean warming as a function of depth and basin[15,18–20], relatively little analysis has been undertaken of the distribution as a function of water-mass layers and within specific water masses. This is the focus of the present study.

[1]Climate Change Research Centre, University of New South Wales, Sydney, NSW 2052, Australia. [2]Australian Centre for Excellence in Antarctic Science, University of New South Wales, Sydney, NSW 2052, Australia. [3]Centre for Marine Science and Innovation (CMSI), University of New South Wales, Sydney, NSW 2052, Australia. [4]NIOZ Royal Netherlands Institute for Sea Research, Department of Ocean Systems, 1790 AB Den Burg, Texel, The Netherlands. ✉e-mail: zhi.li4@unsw.edu.au

Ocean warming is often measured by tracking changes in ocean heat content (OHC). A useful coordinate to analyze OHC changes is a density- or temperature-based framework because this is a more natural coordinate to explore the thermodynamics of ocean warming[21–23]. Deepening of isopycnals is associated with ocean heat uptake, with past work suggesting the maximum deepening can be found in the mode water density range over the Southern Hemisphere extratropics and in the North Atlantic, implying a volumetric increase of mode waters and associated subduction and lateral spreading of heat from high-latitude well-ventilated regions[14,18–20,24]. A temperature-based framework further suggests that about half of the surface heat uptake during 1970–2014 is confined to about a quarter of the ocean's surface area in the subpolar regions, which is in turn capable of exchanging heat with the coldest 90% of the global ocean volume[25].

The above studies have provided insights and identified where we lack observational-based constraints in understanding OHC variability. However, most past studies of OHC changes are depth- or basin-integrated, obscuring the OHC evolution in mode and intermediate waters and the related processes at play. Mode and intermediate waters are distinguished by their properties of density, salinity and low stratification by wintertime ventilation, and their importance in storing and redistributing heat, carbon and oxygen[26–28], yet their heat content changes associated with global ocean warming are yet to be quantified. Furthermore, relatively little attention has been paid to their temperature and volumetric evolution and the associated role of subduction, spreading, and mixing in accumulating oceanic heat, despite observations highlighting a significant upper ocean volumetric increase of mode waters[18] and showing striking local warming signals at depths of 700–2000 m in the North Atlantic and Southern Oceans[19,29].

In this study, we examine the global and regional variability of OHC in the upper 2000 m of the ocean using products of subsurface temperature and OHC that have been constructed from historical measurements and Argo observations. We explore heat accumulation within global mode and intermediate water layers and their role in increasing global OHC. We further define a set of regional mode and intermediate waters using specific geographic and density constraints within those two layers ("Methods"). Using a classification of water masses allows us to quantify observed changes in the heat content of individual mode and intermediate waters and to calculate their contribution to the global OHC increase. We separately consider OHC increases due to volume and temperature changes to better understand the processes driving ocean heat uptake and redistribution by these water masses.

## Results

### Accelerating global ocean warming

Observations of the interior ocean thermal structure relied on measurements from research vessels prior to the advent of expendable bathythermographs (XBTs), which have been widely used since the 1970s. During the 1990s, unprecedented coverage of the global ocean using high-quality hydrographic sections was achieved as part of the World Ocean Circulation Experiment[30] (WOCE). Subsequently, the Argo float array began to dominate ocean observations since around 2005, measuring the upper 2000 m of the water column. These observing systems have evolved to provide a valuable means to improve OHC estimation, leading to major advancements in global climate studies[6]. The community has also made considerable progress in developing gridded products of subsurface temperature for the global ocean by merging all available observing systems[8,10,31,32]. Due to instrumental biases in bathythermograph measurements[33] and limited data coverage prior to Argo (before 2005), as well as geographic and depth limitations of conventional Argo floats, various correction schemes and objective analysis methods have been applied to mitigate these biases and increase our confidence in historical OHC estimates.

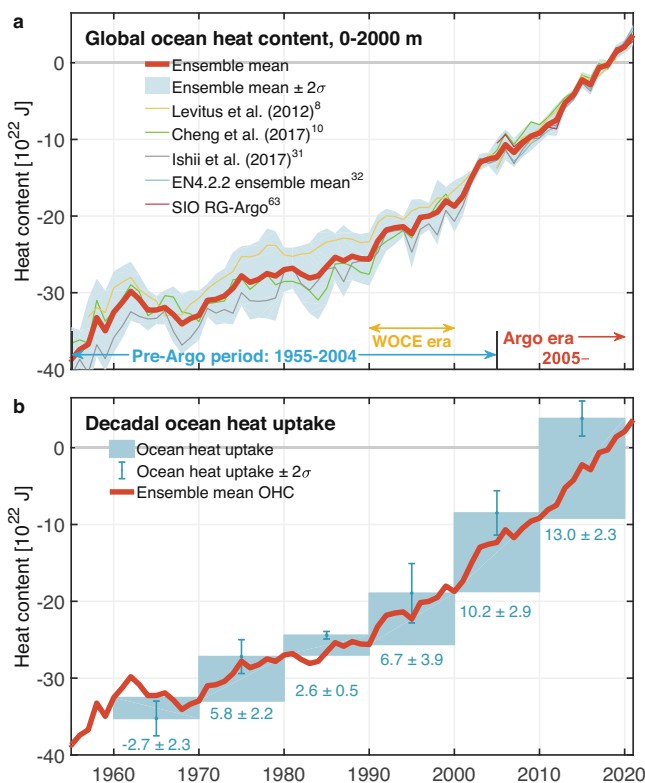

**Fig. 1 | Multi-decadal acceleration in global ocean warming across the measurement era. a** Time series for the heat content ($10^{22}$ J) of the upper 2000 m of the ocean relative to 2016–2020 mean, using various observational products[8,10,31,32,63]. Red lines in panels (**a**) and (**b**) represent the ensemble mean time series of global ocean heat content (OHC) during 1955–2020, and shading in panel (**a**) indicates the $\pm 2$ ensemble standard deviation uncertainty range ($\pm 2\sigma$) for the global OHC time series. **b** Blue rectangle bars indicate the ensemble-averaged global ocean heat uptake ($10^{22}$ J) for every 11-year period across the measurement era ("Methods"). Superimposed error bars indicate the $\pm 2$ ensemble standard deviation uncertainty range ($\pm 2\sigma$) of global ocean heat uptake across various datasets. Pre-Argo period represents the period 1955–2004, WOCE era represents the Hydrographic Program of the World Ocean Circulation Experiment during the 1990s, and Argo era indicates the period with Argo record since 2005.

To provide a more robust analysis of the global and regional OHC changes in this study, we present the ensemble average of OHC estimates derived from various ocean data products (Fig. 1; "Methods"). The uncertainty of this estimate is taken to be $\pm 2$ standard deviations ($\pm 2\sigma$), to indicate the degree of agreement across the products.

Although some differences in the global OHC estimates are evident among the different datasets, the ensemble mean of OHC estimates for the upper 2000 m shows that ocean warming is not only continuing but accelerating[5,16,17] (Fig. 1). The average linear trend of global OHC over the pre-Argo period 1955–2004 is $3.9 \pm 0.7 \times 10^{21}$ J yr$^{-1}$ (Fig. 1a; Table 1). For the Argo era 2005–2020, we estimate this trend to have increased almost threefold to $10.2 \pm 1.8 \times 10^{21}$ J yr$^{-1}$. Further comparison with previous estimates of OHC trends (Table 1) and the latest assessment report of the Intergovernmental Panel on Climate Change (IPCC) Working Group I[7] also reveals significant increases in the ocean warming rate across the measurement record[5,8–12,17,18,20]. This increasing ocean warming is not just limited to the upper layer of the ocean but also occurs at intermediate depths (700–2000 m) according to a further breakdown of OHC trends by depth over the upper 2000 m[5,7].

When evaluating the ocean heat uptake for each decade ("Methods"), analysis of the past three decades reveals that the ocean heat uptake during 2010–2020 has increased more than 25% relative to

**Table 1 | Accelerating global ocean warming, the ensemble mean and ±2σ of ocean heat content (OHC) trends from various observational products versus previous measurements**

| Literatures (e.g.) | OHC trend ($10^{21}$ J yr$^{-1}$) | Period (year) |
|---|---|---|
| This study | 3.9 ± 0.7 | 1955–2004 |
| Levitus et al. (2012)[8] | 4.3 | 1955–2010 |
| Häkkinen et al. (2016)[18] | 4.8 | 1957–2014 |
| Cheng et al. (2017)[10] | 8.4 | 1998–2015 |
| Roemmich et al. (2015)[12]* | 6.3 | 2006–2013 |
| Kolodziejczyk et al. (2019)[20] | 8.0 | 2006–2015 |
| Rathore et al. (2020)[9] | 8.4 | 2005–2015 |
| This study | 10.2 ± 1.8 | 2005–2020 |
| This study | 11.8 ± 2.1 | 2010–2020 |

The OHC trend values are given in the second column in units of $10^{21}$ J yr$^{-1}$, and the third column denotes the period analyzed, with periods quoted inclusive of start and end years.
*The OHC trend of $6.3 \times 10^{21}$ J yr$^{-1}$ is given as the average of OHC trends in Table 1 of ref. 12. Note that to obtain the values of decadal ocean heat uptake shown in Fig. 1b (units of $10^{22}$ J), the average OHC trend shown here (units of $10^{21}$ J yr$^{-1}$) is multiplied by the relevant time period in Fig. 1b (i.e., 11 years for the 2010–2020 period).

2000–2010 and has nearly doubled relative to the 1990's WOCE era, as seen in Fig. 1b, where we highlight the decadal ocean heat uptake since the 1960s. Note that there has been both increased ocean sampling and a shift of the observational network from a ship-based system to the Argo network since the initiation of the global Argo array (2001–2003)[34]. This may impact the estimated increase in global ocean warming over the past three decades (Fig. 1). However, the rate of global mean sea-level rise has also been increasing since 1993 based on an independent estimate from satellite altimeter data[1,35], providing confidence in our results given that half of the global sea surface height increase is due to thermal expansion of the ocean since altimeter measurements began[1]. Significant ocean warming and accelerating OHC changes are also consistent with the increase in net radiative energy absorbed by Earth detected in satellite observations[5,36], something that is likely to continue throughout the 21st century[6,16] in the absence of substantial greenhouse gas emissions reductions.

The increased ocean warming is non-uniformly distributed across ocean basins. Overall, in each ocean basin, an increase in OHC is observed (values indicated in Fig. 2a, b), with stronger warming in the mid-latitude Atlantic Ocean and the Southern Ocean compared with other basins[6,12]. Total warming in the Southern Ocean is estimated to account for ~31% of the global upper 2000-m OHC increase from 1980–2000 to 2000–2010 (Fig. 2a), and almost half of the global OHC increase from 2000–2010 to 2010–2020 (values indicated in parentheses of Fig. 2b). Hence the Southern Ocean has seen the largest increase in heat storage over the past two decades, holding almost the same excess anthropogenic heat as the Atlantic, Pacific, and Indian Oceans north of 30°S combined (Fig. 2d). The most striking warming in the Southern Ocean is concentrated on the northern flank of the Antarctic Circumpolar Current, the location of deep mixed layers and subduction hotspots for Subantarctic Mode Water and Antarctic Intermediate Water, as well as the location of subtropical mode waters formation further equatorward (Fig. 3). The well-ventilated regions near western boundary current extensions in the North Atlantic and North Pacific also reveal large warming over the past two decades. These hotspots of ocean warming are likely linked to enhanced uptake, subduction, and lateral spreading of heat associated with mode and intermediate waters that warrant further investigation.

Even though all products show robust ocean warming rates averaged in the upper 2000 m over the past few decades, reaching record warming over 2010–2020[5,17] (Table 1), when decomposed into water-mass trends these products exhibit better agreement after ~2005 once the Argo network was widely deployed. Thus, for the

remainder of this study, we focus on the Argo era 2005–2020, a period common to all data products analyzed. We also limit the water-mass decomposition to be between 65°S and 65°N, because of sparse observations in the polar regions even after the Argo array was established.

## Heat accumulation in mode and intermediate water layers

The global OHC change of the upper 2000 m is next decomposed into the tropical water layer, the mode water layer, and the intermediate water layer to study the distribution of ocean warming by global water-mass layers (Fig. 4). The most remarkable OHC change over the Argo era is the increase within the mode water layer (Fig. 4c, f). The mode water layer occupies only 20% of the upper 2000 m of the ocean, yet plays a dominant role in total ocean heat content increase (Fig. 4a, Supplementary Table 1). The intermediate water layer occupies 42% of the upper 2000 m, but exhibits a decreasing heat content trend (Fig. 4d). These opposing signals are mainly due to the mode water layer accumulating heat content via increasing its volume, while the intermediate water layer loses heat content due to a volume decrease (Fig. 5c, f). After factoring out volumetric effects, both the global mode and intermediate water layers show a near-continuous and monotonic warming across the Argo era (Fig. 5, Supplementary Fig. 2), revealing a robust footprint of global warming penetrating into the ocean interior. Taken together, the total heat accumulated in the mode and intermediate water layers is 89% of the net global ocean warming during the Argo era, despite their total volume just occupying 62% of the upper 2000 m. The net warming of these two layers, separate from volumetric effects, accounts for ~76% of global ocean warming, while ~12% is due to their combined volume change (Fig. 5a, Supplementary Table 2).

Heat content within the tropical water layer reveals large variations associated with El Niño-Southern Oscillation (ENSO) variability (Figs. 4b, f and 5b). In particular, the heat content of the upper 100 m increases significantly during El Niño years (2009/10, 2014/15, 2015/16, and 2018/19), offset partially by opposing variability below 100 m, confirming previous studies[12,37]. This ENSO-driven variability in OHC is related to net air-sea heat exchanges and the redistribution of heat laterally across different latitudes and ocean regions, and vertically between surface and subsurface layers[37,38]. In this way, tropical warming exhibits strong interannual variability, mainly due to El Niño, whereas warming in global mode and intermediate water layers is less variable and characterized by a near-continuous accumulation of heat. Because of the tremendous heat accumulation within the mode and intermediate water layers, we next focus on the OHC increase in a few key water masses found within these two layers.

## Regional intensification in ocean warming

To further explore how the upper 2000-m ocean warming is distributed by basin during the Argo era 2005–2020, we next compute heat content trends over the upper 2000 m and find the increased OHC to be concentrated in the tropical Pacific, subtropical oceans and the Southern Ocean (Fig. 6a). Note that trends within this relatively short period can indicate both the forcing signal due to anthropogenic warming as well as signals superimposed due to internal climate variability. For example, the OHC change in the tropical Pacific shows an El Niño-like spatial structure, with warming in the east and cooling in the west (Fig. 6a). This is due to the prevalence of El Niño events during the latter half of the analyzed time period, relative to the start of that period, which was characterized by three strong La Niña events (2007/08, 2010/11 and 2011/12). This is linked to a transition of the Inter-decadal Pacific Oscillation (IPO) from its negative to positive phase in around 2014[39] (note that our analysis period does not include the recent triple La Niña spanning 2020–2023). Warming in the extra-tropical latitudes is mainly confined to the subtropical Atlantic and Pacific Oceans in both hemispheres and the Southern Ocean. From the

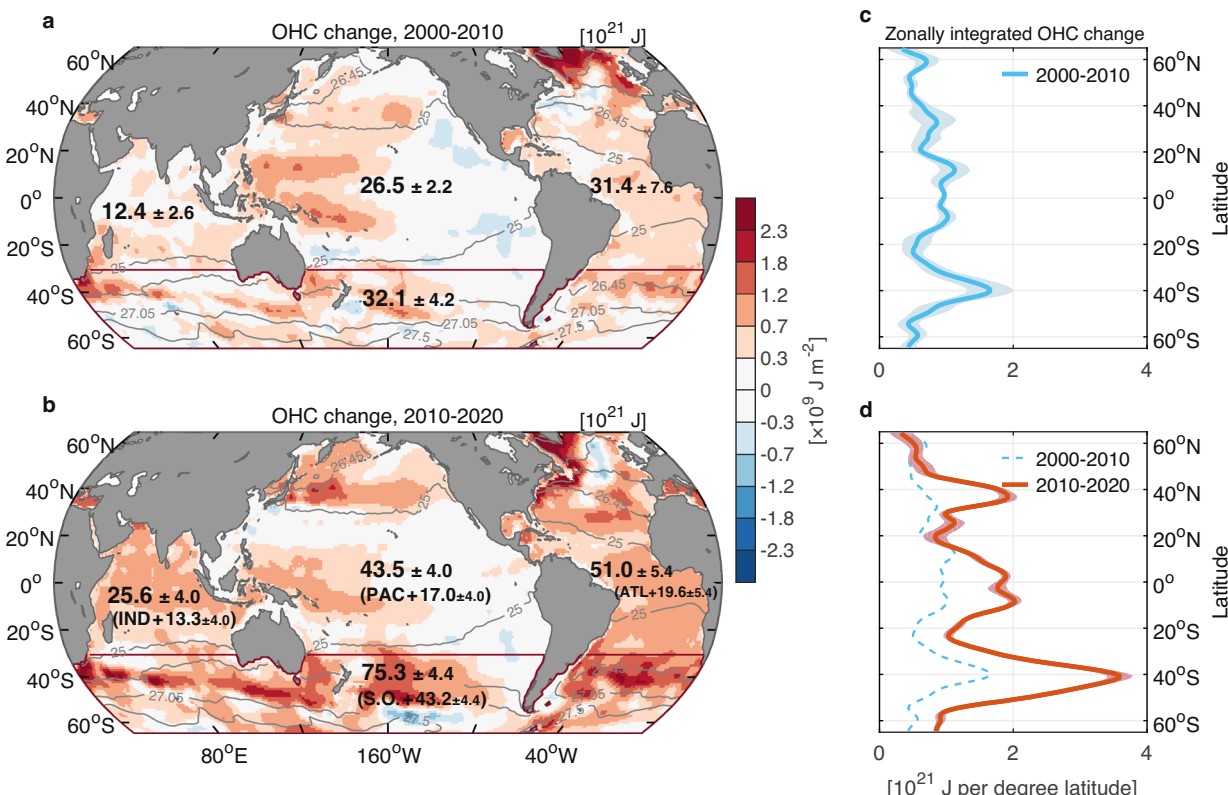

**Fig. 2 | Regional intensification in ocean warming over the past two decades, 0–2000 m.** The ensemble mean of ocean heat content (OHC) changes averaged for years **a** 2000–2010 and **b** 2010–2020, relative to the 1980–2000 mean. Units of shadings in panels (**a**, **b**) are shown as $10^9$ J m$^{-2}$. The values over each basin indicate the OHC increase relative to the 1980–2000 mean over the Southern (S.O., south of 30°S, dark-red line), Atlantic (ATL), Pacific (PAC), and Indian (IND) Oceans, and are limited to 65°S–65°N. Units are shown as $10^{21}$ J. The values in parentheses in panel (**b**) indicate the basin-integrated OHC increase from 2000–2010 to 2010–2020. The basin mask used to distinguish ocean basins of the Southern, Atlantic, Pacific, and Indian Oceans is obtained from ref. 8. Superimposed gray contours represent the positions of wintertime isopycnals $\gamma^n$ = 25, 26.45, 27.05, and 27.5 kg m$^{-3}$ at 10 m depth from SIO RG-Argo. **c**, **d** Zonally integrated OHC change ($10^{21}$ J per degree latitude) versus latitude for the period 2000–2010 (blue line), and 2010–2020 (red line), relative to the 1980–2000 mean. Lines in panels (**c**) and (**d**) represent the ensemble mean, and shadings indicate the ±2 ensemble standard deviation uncertainty range (±2$\sigma$) of OHC changes.

zonally integrated OHC increase, we find that this extratropical ocean warming can be observed vertically from the surface down to 300–1500 m depth, covering the mode and intermediate water layers (Fig. 7).

An independent measure of OHC changes can be found in altimeter measurements of sea level if salinity adjustments remain small and the ocean mass component of sea-level change is known and accounted for. This is because the ocean expands when it is warmed, thus directly affecting sea level. We find that the locations of strong increase in OHC and thermal expansion (Fig. 6a, c) correspond closely to regions where the sea level has also increased[40,41] (Fig. 6b). These locations include the eastern tropical Pacific, the subtropical oceans, and the Southern Ocean. The effects of changed ocean salinity are small (Fig. 6d), mainly limited to the North Atlantic where Greenland ice melt and adjustments in the Atlantic Meridional Overturning Circulation have had an impact[42,43]. The mass component of sea-level change, due to land ice melt (including glaciers, and the Greenland and Antarctic ice sheets) and land water storage change, has also significantly increased across the altimeter era, but displays a more uniform pattern of sea-level rise compared to the thermal expansion component[40,41].

### Increased heat uptake by mode and intermediate waters
Anthropogenic warming has stratified most regions of the world's oceans[44], thereby, reducing ocean ventilation and slowing vertical exchanges of heat and carbon. However, it has been estimated that the total ocean from the surface to the seafloor has been warming at an

accelerating rate[11] (Fig. 1), and there is an indication that mode waters, at least on the basin average, have increased in their volume and depth[14,18–20,24,45]. Here we show that the accelerated OHC increase is concentrated in the subtropical and Southern Ocean (Figs. 2 and 6), from the surface to 300–1500 m depth (Fig. 7), spanning the mode and intermediate layers (Figs. 4, 5 and 7). These regions are characterized by deep wintertime mixed layers and vigorous mode water formation[46] (Fig. 3), which act to imprint a signature of air-sea interaction and climate conditions in the ocean interior. Therefore, a component of this accelerated OHC increase must be due to increased heat uptake by mode and intermediate waters via wintertime ventilation, in combination with deepening of the warmed water by mixing. In this way warming from the surface can propagate into the ocean interior and spread laterally and vertically via circulation and mixing with surrounding water masses. This motivates us to study the heat content changes within individual mode and intermediate waters to better understand their role in warming the ocean. We do this by first defining 13 regional water masses using density and geographic limits within the mode and intermediate layers and over the subtropical and subpolar oceans respectively ("Methods"; Figs. 3 and 7, Supplementary Table 3), and then estimating their OHC changes over the Argo era 2005–2020 (Fig. 8).

The total heat content of a water mass can increase because it warms, or because the total volume of that water mass increases, or both ("Methods"). By decomposing the OHC increase over the Argo era 2005–2020, we find that the actual warming (within a fixed volume) of all regionally defined mode and intermediate waters can

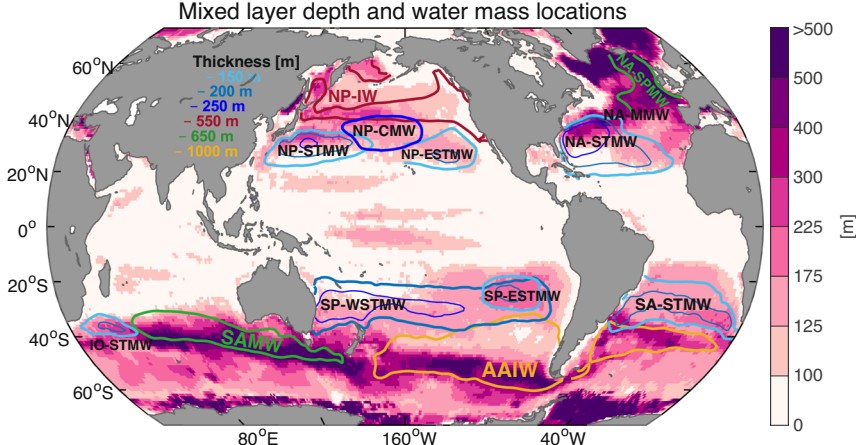

**Fig. 3 | Maximum mixed layer depth (MLD) across the Argo era and locations of mode and intermediate waters in the global ocean.** MLD is color shaded while water masses are indicated by name and contours for thickness. The MLD is defined as the depth at which density is 0.03 kg m$^{-3}$ greater than the value of density at 10-m depth from SIO RG-Argo, IAP data, and EN4.2.2 ensemble mean. Superimposed acronyms represent Subantarctic Mode Water (SAMW), Antarctic Intermediate Water (AAIW), North Atlantic STMW (NA-STMW), North Atlantic Madeira Mode Water (NA-MMW), North Atlantic SPMW (NA-SPMW), South Atlantic STMW (SA-STMW), North Pacific STMW (NP-STMW), North Pacific Eastern STMW (NP-ESTMW), North Pacific Central Mode Water (NP-CMW), North Pacific Intermediate Water (NP-IW), South Pacific Western STMW (SP-WSTMW), South Pacific Eastern STMW (SP-ESTMW), Indian Ocean STMW (IO-STMW). The density and geographic constraints for defining these mode and intermediate waters are detailed in "Methods" and Supplementary Table 3. Superimposed contours represent the 150-m, 200-m, and 250-m thickness of all STMWs (blue), 550-m thickness of NP-IW (red), 650-m thickness of SAMW and NA-SPMW (green), and 1000-m thickness of AAIW (yellow), to indicate their geographic locations. Note that the water-mass thickness is estimated based on the density constraints given in Supplementary Table 3, and for simplicity, only the core thickness of each water mass is shown. The geographic constraints for estimating heat content change of water masses are referred to in Supplementary Table 3. Further overview of the mode and inter-mediate water subduction sites can be found in refs. 26,28,46–49,54–56,61.

account for 48% of the global ocean warming (Figs. 8e and 9g, Supplementary Tables 3 and 4). The volume of each of these water masses also changes markedly but non-uniformly across this period, with their combined volume change modestly offsetting (−4%) the net mode and intermediate water warming calculated over a fixed volume. The combined OHC increase of the sum of all these regionally defined mode and intermediate waters, due to both warming (fixed isopycnal depth and volume) and volume changes, amounts to a total contribution of 44% to the global OHC increase (using the values shown in Fig. 8e and Supplementary Table 4). The total volume of the regionally defined water masses within the subtropical and subpolar oceans occupies just 24% of the upper 2000 m averaged over 2005–2020. Therefore, this disproportionately large heat uptake per volume of mode and intermediate waters reveals their outsized role in absorbing and storing heat for the world ocean under global warming. We next separately consider the OHC increase due to both volume and temperature changes.

**Warming and thickening of water masses**

The most striking ocean warming and thickening of the mode and intermediate water layers is over the subtropical Atlantic and Pacific Oceans and in the Southern Ocean, dominating the global OHC increase (Figs. 5, 7, 8 and 9). The Subtropical Mode Water layer has shown the largest thickening of the upper 2000-m ocean over the Argo era (Fig. 5, Supplementary Fig. 2). This thickening and warming of the layer are concentrated mainly over the subtropical Atlantic and Pacific Oceans, with a weak increase over the subtropical South Indian Ocean (Figs. 8b and 9a, d), which likely corresponds to ventilation regions of regional Subtropical Mode Waters[26,47] (Fig. 3). The Subpolar Mode Water layer and the intermediate water layer both show broad warming (volume effects removed) over the Southern Ocean (Fig. 9e, f). In the Southern Ocean the 0–1500-m depth range is occupied by Subantarctic Mode Water and Antarctic Intermediate Water[26,48,49]. Warming from the sea surface to 800 m has been measured along a single transect in the Southern Ocean across mode and intermediate water layers[29]; this can also be seen in our analysis of the full zonally integrated OHC trends (Fig. 7). In particular, our analysis reveals that

Subantarctic Mode Water has warmed and deepened substantially within and north of its ventilation region in the Southern Ocean, while Antarctic Intermediate Water has warmed and thinned due to the deepening of lighter isopycnals (Figs. 8c, d and 9e, f, Supplementary Table 4). Total warming (volume effects removed) of Subantarctic Mode Water and Antarctic Intermediate Water accounts for ~36% of global ocean warming over the Argo era (Fig. 9g).

Among the mode and intermediate waters defined in this study, we find that OHC trends of mode waters are dominated by their volumetric changes. Instead warming (fixed volume) exhibits a larger signal in the heat content trend for intermediate waters, including both Antarctic and North Pacific Intermediate Water (Figs. 3 and 9, Supplementary Fig. 2 and Table 4). This is likely because while the warming of water masses propagates downward, the density-defined boundary of water masses shifts both vertically and laterally as a response to ocean warming (Supplementary Fig. 3). In particular, the subtropical oceans and the Southern Ocean reveal a significant deepening and poleward shift of mode water isopycnals. This indicates that the Subtropical/Subantarctic Mode Water has been expanding and shifting toward locations that once were classified as Subpolar Mode Water and/or intermediate water. Note that although the mode water has shifted, this does not necessarily indicate a net warming of the water mass. Instead, the intermediate water layer could potentially be lightened and warmed due to mixing with the warmed and deepened mode waters over the subtropical oceans and the Southern Ocean[19] (Fig. 9). In reality, a combination of these two effects has likely played out. In addition to density-based constraints, alternative criteria like stratification or salinity within specific locations are often used to define mode and intermediate waters[26]. These criteria also allow the examination of changes in the density and temperature of water masses. Within this framework, the poleward and downward migration of isopycnals over the subtropical oceans and the Southern Ocean can serve as an indication of the warming and lightening of mode and intermediate waters (Supplementary Fig. 3).

There are several possible mechanisms driving the increased ocean heat uptake, particularly over the ventilation regions of mode

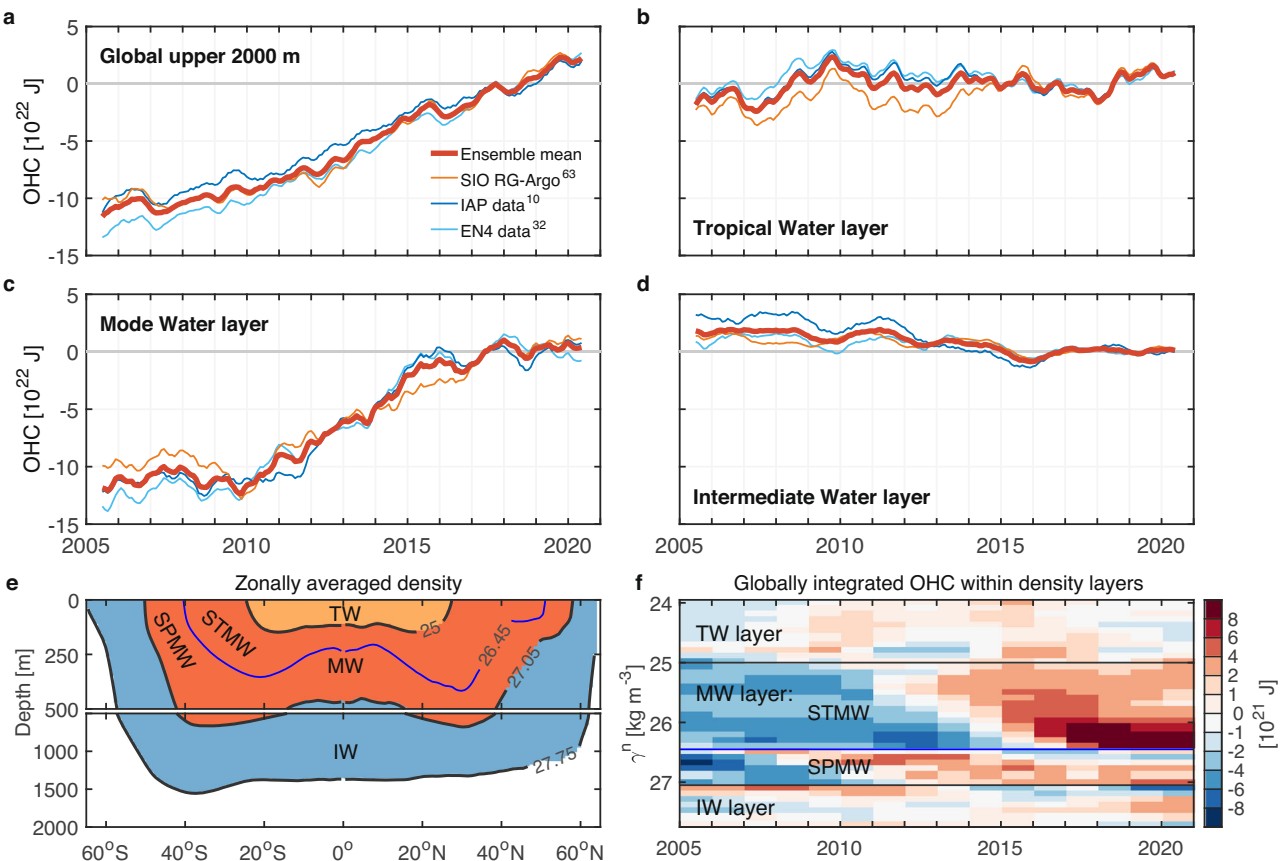

**Fig. 4 | Global ocean heat content (OHC) change relative to the 2016–2020 mean, shown both globally and also decomposed into key water mass layers.** Solid lines are the 13-month running means ($10^{22}$ J). **a** Upper 2000 m of the ocean, 65°S–65°N. Thin lines indicate the OHC change from SIO RG-Argo (yellow), IAP data (blue), and EN4.2.2 ensemble mean (light blue), respectively. The thick line indicates the ensemble mean time series. **b** The tropical water layer. **c** The mode water layer. **d** The intermediate water layer. **e** Zonally averaged density to indicate the geographic locations of the tropical water layer (TW, yellow shading), the mode water layer (MW; red shading) and its components within the Subtropical Mode

Water layer (STMW) and the Subpolar Mode Water layer (SPMW), and the intermediate water layer (IW, blue shading) over the upper 2000 m of the ocean ("Methods"). Superimposed contours represent the wintertime isopycnals of $\gamma^n = 25, 26.45, 27.05$ and $27.75$ kg m$^{-3}$ from SIO RG-Argo. **f** Globally integrated annual mean OHC distributed by density and time ($10^{21}$ J per 0.1 kg m$^{-3}$ density bin), with the averaged OHC during the Argo era being removed and then the 3-year running mean being applied to every layer. Colors indicate the ensemble mean estimate of OHC changes from SIO RG-Argo, IAP data, and EN4.2.2 ensemble mean.

and intermediate waters. Anthropogenic warming provides the excess heat to warm the ocean in regions where sea surface temperature is cooler than the overlying surface air temperature—for example, an acceleration and poleward shift of the westerly winds[50] has kept surface waters relatively cool in the Southern Ocean, and thus favorable for ongoing ocean heat uptake[51]. The circulation-related changes in heat transport also play a profound role in heat accumulation over the subtropical Atlantic Ocean and the Southern Ocean[15,52]. In addition, western boundary currents accumulate and feed heat toward the formation regions of Subtropical Mode Waters. Deep mixed layers at the mode water formation sites pump heat into the ocean interior via subduction[53–55] (Fig. 3). In the Southern Ocean, strengthening westerly winds and Ekman pumping steepen isopycnal surfaces and increase the Subantarctic Mode Water volume via subduction[56,57], while also pushing cold surface Antarctic water northward[58], enabling increased heat uptake and subduction into mode and intermediate layers[51], as noted above. Enhanced mesoscale eddy activity over the subtropical oceans and the Southern Ocean[59] might also contribute to increased subduction and vertical transport of heat for Subtropical and Subantarctic Mode Waters and Antarctic Intermediate Water[54,60,61].

While the globally integrated OHC of the upper 2000 m and in the mode and intermediate water layers both show a robust and near-monotonic increase (Figs. 1 and 5), variations in the formation mechanisms, properties, and heat content of localized mode and intermediate waters can be influenced by interannual to decadal variability (e.g., ENSO and the Indian Ocean Dipole[62] on interannual time-scales, and the IPO on decadal time-scales). However, our analyses show that the main footprint of internal climate variability occurs over the tropical Pacific Ocean, whereas the rest of the Pacific as well as the Indian, Atlantic, and Southern Oceans all tend to show a consistent and robust warming, in particular over the subduction and well-ventilated regions near western boundary current extensions and north of the ACC (compare Figs. 2a, b and 6a). The near-monotonic warming in mode and intermediate water layers across the Argo era (Figs. 4 and 5a), suggests a robust ocean heat gain forced by anthropogenic climate change. In contrast, OHC changes within the tropical water layer and in the tropical Pacific Ocean reveal variations likely associated with ENSO and/or IPO variability.

## Summary and discussion
The world's oceans hold by far the largest excess anthropogenic heat in Earth's climate system[1]. Knowing exactly where and how much heat is stored in the ocean is thus fundamental to our ability to understand and predict future climate change and sea-level rise. In addition, this knowledge can help inform how to optimize observing networks for monitoring the state of the ocean and climate system. Our study has documented a strong acceleration in global ocean warming since the 1990s, amounting to >25% increase in OHC during

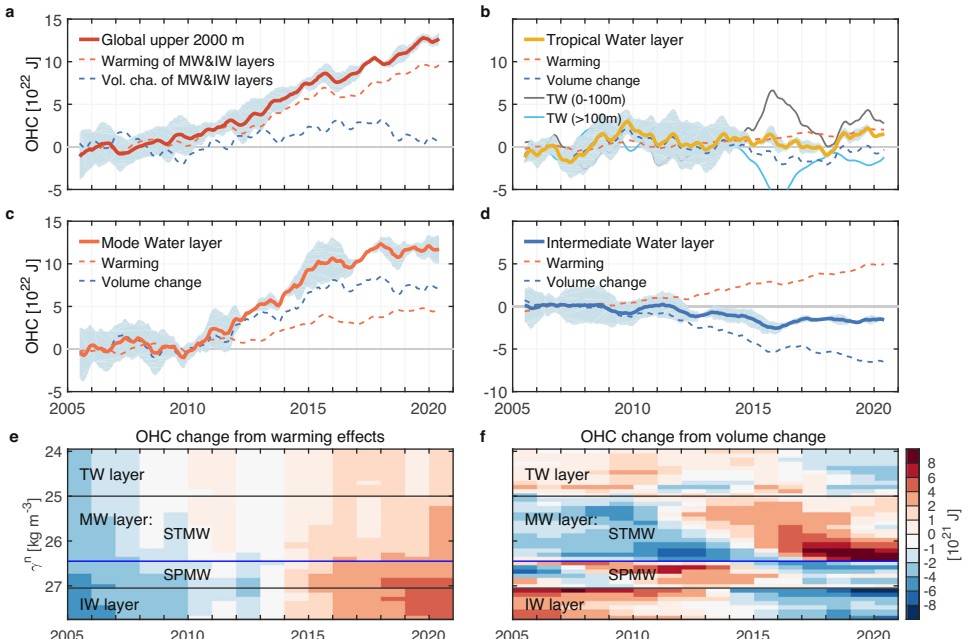

**Fig. 5 | Decomposition of ocean heat content (OHC) change into warming and volume change components.** Note that for clarity of this decomposition, the values are shown relative to the 2005–2009 ensemble mean. Lines are the 13-month running means ($10^{22}$ J). **a** Upper 2000 m of the ocean between 65°S–65°N. The bold red line represents the ensemble mean time series from SIO RG-Argo, IAP data, and EN4.2.2 ensemble mean, and shading indicates the ±2 ensemble standard deviation uncertainty range (±2σ). Red and blue dashed lines indicate the warming and volumetric components of OHC changes in the global mode and intermediate water layers, respectively. **b** The tropical water layer (bold yellow line) and its components within the upper 100 m (gray line) and in the subsurface layer below 100 m (light blue line), as well as its warming (red dashed line) and volumetric change (blue dashed line) components. **c** The mode water layer, and its warming (red dashed line) and volumetric change (blue dashed line) components. **d** The intermediate water layer, and its warming (red dashed line) and volumetric change (blue dashed line) components. **e, f** The warming (**e**) and volumetric change (**f**) components of globally integrated OHC distributed by density and time ($10^{21}$ J per 0.1 kg m$^{-3}$ density bin), with the averaged OHC during the Argo era being removed and then the 3-year running mean being proceeded to every layer. Colors indicate the ensemble average of OHC changes from SIO RG-Argo, IAP data, and EN4.2.2 ensemble mean.

2010–2020 relative to 2000–2010, and nearly a twofold increase during 2010–2020 relative to 1990–2000. Approximately 89% of the increase in global ocean heat uptake, which includes warming and volume changes, is confined to global mode and intermediate water density layers during the Argo era 2005–2020. Mode and intermediate waters play a vital role in absorbing and redistributing tracers such as heat, carbon, and oxygen in the global ocean. The warming of 13 of these regionally defined mode and intermediate waters at fixed geographic locations and isopycnal depths (i.e., with volume effects removed) accounts for ~48% of global ocean warming despite their volume occupying just 24% of the ocean. This includes Subtropical Mode Waters in both hemispheres, which have overall shown a striking increase in OHC dominated by increases in volume, and Subantarctic Mode Water and Antarctic Intermediate Water in the Southern Ocean, which have warmed substantially and are responsible for more than one-third of global ocean warming across the 2005–2020 period.

Mode and intermediate waters are traditionally defined by their oceanographic features such as temperature, density, salinity, and stratification. These features have evolved as the upper ocean has warmed and stratified over the last few decades, and as precipitation minus evaporation fields have changed. Here for simplicity, we use a set of fixed longitude, latitude, and density or depth criteria to define these water masses and estimate how their OHC has changed over the Argo era, although we also separate out the effects of volumetric changes from water-mass transformation in attributing total OHC change to each water mass. Our analysis of heat accumulation within these key water masses focused on the Argo era 2005–2020 when the subsurface thermal structure of the ocean was well-constrained by the Argo network. Although our results are robust across a set of

observational-based datasets, important questions remain, such as separating out the anthropogenic signal of OHC change from internal climate variability, particularly in mode and intermediate waters where decadal variability is poorly constrained by sparse measurements prior to the Argo era. In addition, due to sparse data coverage over the ice-covered oceans, in marginal seas, in the deep and abyssal oceans, and in the years prior to the Argo network, not all regions of the ocean or time scales are well-constrained by a suitably dense array of measurements. It is thus essential to continue establishing and improving coastal, polar, and deep ocean observing systems to better monitor changes in the ocean and track the global energy balance more reliably.

Our work reveals accelerated warming in the upper 2000 m of the ocean over the past several decades and highlights increasing heat uptake by mode and intermediate waters, with these two water masses responsible for the majority of ocean warming over the Argo era (2005–2020), despite a limited area of interaction with the atmosphere. Exactly how this heat uptake plays out over the coming decades and beyond remains highly uncertain. For example, climate change-induced warming and freshening at the surface are projected to stratify the upper ocean, which will reduce the overturning of these water masses, in turn reducing their capacity to uptake heat. This would have profound implications for the rate of future anthropogenic climate change.

## Methods
### Observational products
A total of 5 state-of-the-art grided ocean datasets and OHC products are used in this study to estimate global ocean warming rates and three of those products are used to estimate OHC changes within the global

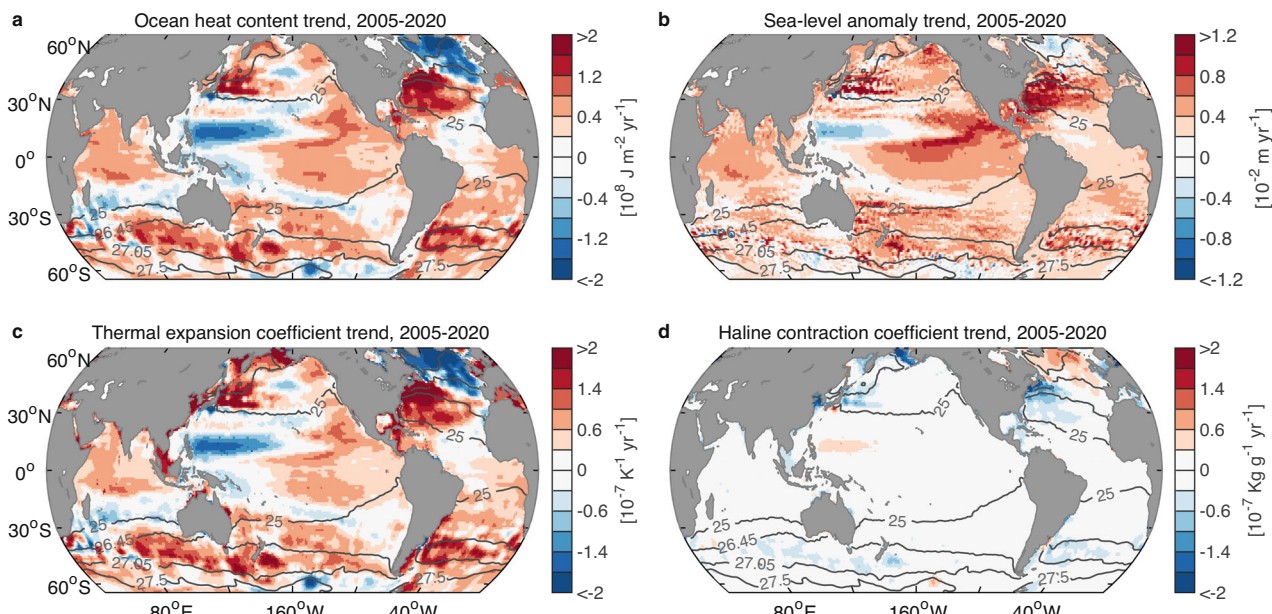

**Fig. 6 | Temporal linear trends in ocean heat content, sea-level anomaly, and thermal expansion and haline contraction coefficients over the Argo era 2005–2020. a** Heat content trend ($10^8$ J m$^{-2}$ yr$^{-1}$). **b** Sea-level anomaly trend ($10^{-2}$ m yr$^{-1}$). Trends in vertically averaged **c** ocean thermal expansion coefficient ($10^{-7}$ K$^{-1}$ yr$^{-1}$) and **d** haline contraction coefficient ($10^{-7}$ Kg g$^{-1}$ yr$^{-1}$) over the upper 2000 m of the ocean. For ease of comparison across panels, the haline contraction coefficient is shown with sign convention reversed: i.e., positive values in red indicate a salinity-

related expansion of the water column. The results presented in panels (**a**, **c**, **d**) represent the ensemble means from SIO RG-Argo, IAP data, and EN4.2.2 ensemble mean. Gray lines represent the positions of wintertime isopycnals $\gamma^n = 25$, 26.45, 27.05, and 27.5 kg m$^{-3}$ at 10 m depth from SIO RG-Argo. Note that extending the analysis period to 2005–2022 shows overall robust trends over the subtropical Atlantic and Pacific Oceans in both hemispheres and the Southern Ocean, as in panels (**a**–**d**).

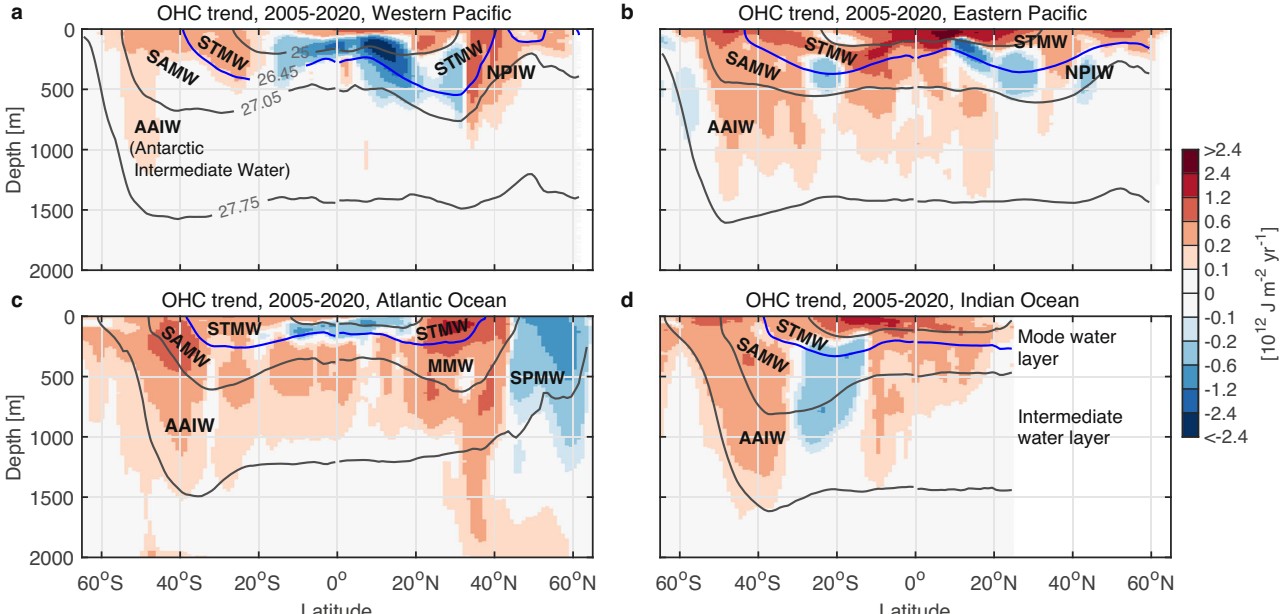

**Fig. 7 | Linear trend in ocean heat content (OHC) zonally integrated over ocean basins.** Units are shown as $10^{12}$ J m$^{-2}$ yr$^{-1}$. **a** Western Pacific Ocean. **b** Eastern Pacific Ocean. **c** Atlantic Ocean. **d** Indian Ocean. The Western and Eastern Pacific Oceans are divided by the longitude of 170°W. The basin mask used to distinguish ocean

basins of the Pacific, Atlantic, and Indian Oceans is obtained from ref. 8. The results presented in panels (**a**–**d**) represent the ensemble means from SIO RG-Argo, IAP data, and EN4.2.2 ensemble mean. Superimposed contours represent the wintertime isopycnals of $\gamma^n = 25$, 26.45, 27.05 and 27.75 kg m$^{-3}$ from SIO RG-Argo.

water-mass layers as well as in regionally defined mode and intermediate waters. These datasets can be grouped into two different types: (1) gridded ocean datasets based on Argo only[63] and (2) four products merged from Argo and other ocean measurements[8,10,31,32,64]. The estimate of global and basin-wide OHC trends among these

selected observational-based products was previously shown to exhibit good agreement in the upper 2000-m ocean over the Argo era[45] (Fig. 1). Several factors can affect the estimate of ocean warming rate and should be factored in, particularly over the pre-Argo period (Fig. 1, Supplementary Fig. 1). Such factors include uncertainties due to data

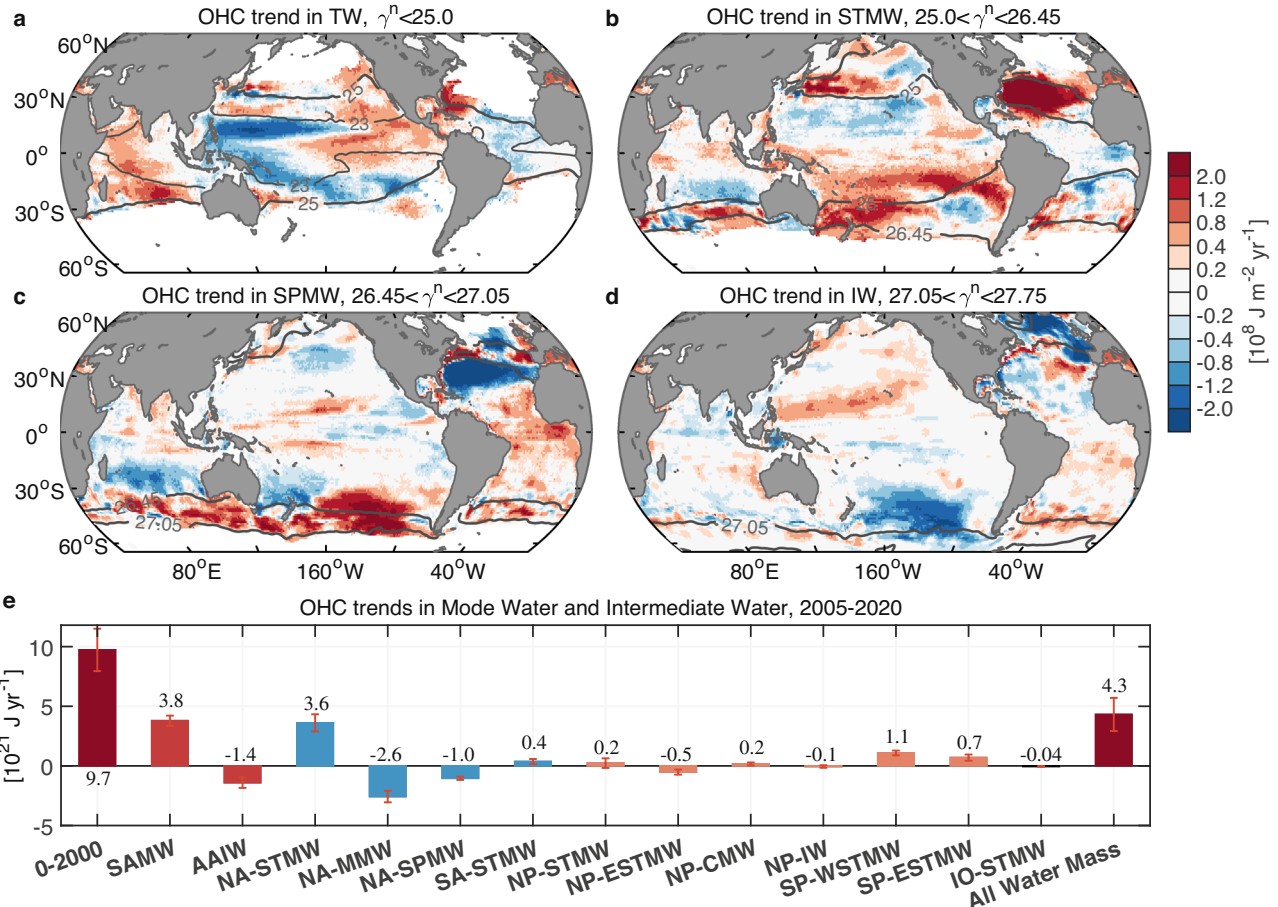

**Fig. 8 | Heat content change in mode and intermediate waters. a–d** The ensemble mean ocean heat content (OHC) trends over 2005–2020 ($10^8$ J m$^{-2}$ yr$^{-1}$) for the **a** tropical water layer, **b** Subtropical Mode Water layer, **c** Subpolar Mode Water layer, and **d** intermediate water layer. Superimposed dark gray contours represent the wintertime isopycnals of $\gamma^n = 23$, 25, 26.45, 27.05 and 27.75 kg m$^{-3}$ from SIO RG-Argo. **e** Trends in OHC ($10^{21}$ J yr$^{-1}$) integrated within the specific water masses; from left to right are the 0–2000 m layer of the world ocean, SAMW, AAIW, NA-STMW, NA-MMW, NA-SPMW, SA-STMW, NP-STMW, NP-ESTMW, NP-CMW, NP-

IW, SP-WSTMW, SP-ESTMW, IO-STMW, and finally the net of all these regionally defined mode and intermediate waters. The density and geographic constraints for defining these mode and intermediate waters are detailed in "Methods" and Supplementary Table 3, and depicted in Fig. 3. Bars in panel (**e**) represent the ensemble average of OHC trends from SIO RG-Argo, IAP data, and EN4.2.2 ensemble mean, and superimposed error bars indicate the ±2 ensemble standard deviation uncertainty range (±2$\sigma$).

scarcity and data quality prior to the late 1960s[10], mapping artifacts from horizontal and vertical interpolation of the sparse historical data[65], the increased spatial sampling coverage over time (particularly in the tropical and Southern Hemisphere oceans), and the shift of the observational network from a ship-based system to the Argo network during 2001–2003[34].

We use monthly means of in-situ temperature and practical salinity from a gridded Scripps Institution of Oceanography (SIO) Argo product[63] (hereafter referred to as "SIO RG-Argo"), which is provided on 56 standard pressure levels (0–2000 dbar) and a 1° regular grid. This gridded Argo product was constructed from Argo data only. We only use the SIO RG-Argo data between 65°S–65°N because of low observational data coverage in high-latitude regions, particularly ice-covered areas and in marginal seas. As such, the resultant global OHC trends from SIO RG-Argo in Fig. 1 will be underestimated if these excluded polar ocean regions are also warming[5]. For the group of gridded datasets merged from both Argo and other historical observations, we include in-situ temperature and salinity observations from the Institute of Atmospheric Physics (IAP) data[10,64] (monthly averaged from 1955 to 2020 with 1° horizontal resolution; referred to as "IAP data") and four ensemble members of EN4.2.2 data from the Met Office Hadley Center[32] (monthly averaged from 1955 to 2020 with 1° horizontal resolution). The EN4.2.2 data used in this study includes all four

ensemble members available from the latest version of the EN4 product, with four different sets of XBT and mechanical bathythermograph (MBT) bias corrections applied (see ref. 32 for more details). When presenting results using the EN4.2.2 data, we choose to show the ensemble average from those four EN4.2.2 ensembles (referred to as the "EN4.2.2 ensemble mean"), to mitigate the uncertainty arising from different approaches applied in correcting biases in XBT and MBT data (Supplementary Fig. 1). Due to the large inconsistency within the four EN4.2.2 ensembles over the 1990s, and due to a large shift of global OHC around 2000–2004 in the EN4.2.2 ensemble mean, we have only included the EN4 data over the Argo period 2005–2020 in this study. The Gibbs-Sea Water (GSW) Oceanographic Toolbox from the TEOS-10 software[66,67] is applied to convert the SIO RG-Argo data and EN4.2.2 data to Conservative Temperature[68] and Absolute Salinity[69], as well as the IAP in-situ temperature to Conservative Temperature, and achieve static stability on time scales of months by applying vertical stabilization software[70]. The thermal expansion and haline contraction coefficients, the surface-referenced potential temperature, $\theta$, and potential density, $\rho$, are then calculated for SIO RG-Argo, IAP data and EN4.2.2 data, using functions from the GSW Oceanographic Toolbox. The neutral density[71], $\gamma^n$, is calculated using functions from PreTEOS-10 software.

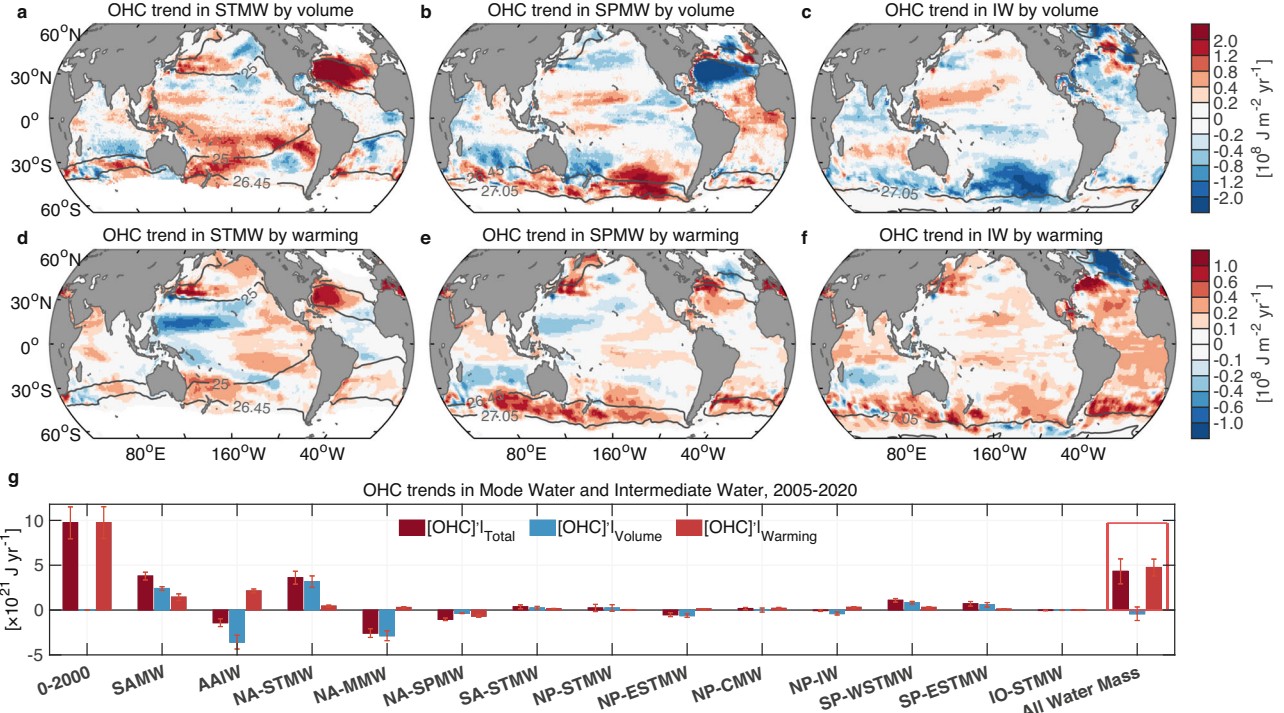

**Fig. 9 | Heat content change due to volumetric increase and warming for 2005–2020.** Trend in ocean heat content (OHC) due to thickness change for the **a** Subtropical Mode Water layer, **b** Subpolar Mode Water layer, and (**c**) intermediate water layer. Units are shown as $10^8$ J m$^{-2}$ yr$^{-1}$. **d–f** Same as (**a–c**) but for the OHC trend due to temperature change. The results presented in panels (**a–f**) indicate the ensemble means from SIO RG-Argo, IAP data, and EN4.2.2 ensemble mean. **g** Same as Fig. 8e but for the OHC trend ($10^{21}$ J yr$^{-1}$, dark red) and its components by volumetric change (blue) and temperature change (orange-red) (detailed in Supplementary Table 4). The locations of the various water masses shown in panel (**g**) are depicted in Fig. 3. Bars in panel (**g**) represent the ensemble average of OHC trends from SIO RG-Argo, IAP data, and EN4.2.2 ensemble mean, and superimposed error bars indicate the ±2 ensemble standard deviation uncertainty range (±2$\sigma$).

The 5-year running mean of global OHC during 1955–2020 from ref. 8 and the annual average of OHC during 2005–2020, updated by the NOAA/NESDIS/NCEI Ocean Climate Laboratory using temperature and salinity fields from the World Ocean Database (both are referred to as "Levitus et al. (2012)"), as well as the annual average of OHC during 1955–2020 from ref. 10 (referred to as "Cheng et al. (2017)") and ref. 31 (referred to as "Ishii et al. (2017)") are applied to reconcile the OHC estimate from SIO RG-Argo and EN4.2.2 ensemble mean in Figs. 1 and 2. These three OHC fields are all provided in a single depth layer as they were vertically integrated over the upper 2000 m. As for the estimate of OHC changes within the mode and intermediate layers and in key water masses in Figs. 3–9 and Supplementary Tables 1–4, and the estimate of ocean thermal expansion and haline contraction in Fig. 6, in which the temperature and salinity fields in the longitude, latitude, and vertical directions are needed, we show the ensemble average of OHC estimates from SIO RG-Argo, IAP data, and EN4.2.2 ensemble mean. The uncertainty range ($\pm 2\sigma$) is derived using the ensemble standard deviation ($\sigma$) of the corresponding estimates to indicate the degree of agreement across the products.

The basin mask used to distinguish ocean basins of the Southern Ocean (south of 30°S), the Atlantic, Pacific, and Indian Oceans in Figs. 2 and 7 is obtained from ref. 8. We also consider changes in sea-level anomaly in Fig. 6, using the E.U. Copernicus Marine Service Information (CMEMS) gridded multi-mission altimeter sea-level anomaly data with 1/4 × 1/4° spatial resolution, covering the Argo era (2005–2020).

## OHC computation and decomposition
This study focuses on the warming of the upper 2000 m of the ocean in part because this is the volume of the ocean in closest contact with the

atmosphere to experience global warming. This 2000-m constraint is also due to the depth limit of the Argo profiling floats, with only sparse data coverage deeper than 2000 m. Waters deeper than 2000 m are generally not ventilated rapidly enough to have experienced the last 50 years of global warming, but note that both the abyssal overturning slowdown[72,73] and oceanic internal waves[74] can result in bottom ocean warming over these time scales, as seen in observations[75].

From the observational-based ocean datasets, we compute the monthly and annual mean OHC (J m$^{-2}$) across the measurement period 1955–2020 for each isopycnal layer as follows:

$$\text{OHC} = C_p \int_{z_1(x,y,z,t)}^{z_2(x,y,z,t)} \rho(x,y,z,t)\,\theta(x,y,z,t)\,dz, \qquad (1)$$

where $C_p$ = 3992 J kg$^{-1}$ K$^{-1}$ is the specific heat capacity of seawater, $\theta$ is the potential temperature with respect to 0 °C, and $z_1$ and $z_2$ (m) are the depth of the lower and upper isopycnal surfaces for integration, respectively. The OHC is partitioned into $\Delta\gamma^n$ = 0.1 kg m$^{-3}$ density bins, spanning between 20 < $\gamma^n$ < 28 kg m$^{-3}$. The OHC trend located at layers denser than $\gamma^n$ > 27.75 kg m$^{-3}$ is not discussed because these layers are not fully sampled within the 2000 m-depth Argo profiles, which also precludes us from producing analyses for deeper depths.

Since the OHC represents the vertical integration of temperature for each isopycnal layer, we separate the temporal change in OHC into two contribution terms[24]:

$$\frac{\partial(\text{OHC})}{\partial t} = \underbrace{C_p \int_{z_1(x,y,z,t)}^{z_2(x,y,z,t)} \frac{\partial \rho(x,y,z,t)}{\partial t}\bar{\theta}(x,y,z)\,dz}_{\text{Thickness Change}} + \underbrace{C_p \int_{\bar{z_1}(x,y,z)}^{\bar{z_2}(x,y,z)} \bar{\rho}(x,y,z)\frac{\partial \theta(x,y,z,t)}{\partial t}\,dz}_{\text{Temperature Change}}$$

$$(2)$$

where $\bar{\rho}$ and $\bar{\theta}$ are the climatologically averaged potential density and temperature over the Argo era 2005–2020 respectively, and the last two terms represent changes in OHC associated with changes in thickness and temperature over the climatologically fixed temperature and density field, respectively. The terms $\bar{z}_1$ and $\bar{z}_2$ indicate the fixed location and depth/thickness of a density layer over the Argo era, based on the climatologically averaged density field. Thus the warming component of OHC changes is estimated based on fixed (Argo era averaged) geographic locations and depths of isopycnals, with a constant volume being used to integrate the warming effects of each water-mass layer and regionally defined water mass over the Argo period.

Note that the calculation of OHC is relative to a fixed reference temperature, making the estimate of trends in OHC and its thickness dependent on the somewhat arbitrary choice of reference temperature when the volume of a water mass changes[76]. In contrast, the temperature component of OHC change is not dependent on the choice of reference temperature. In the calculations presented here, we take the reference temperature to be 0 °C for the global ocean, following similar choices made in past work[20,24]. We also repeated our calculations of OHC trends using −2 °C (the approximate freezing point of seawater) as the fixed reference temperature and found the OHC trends of water masses to be overall robust to this choice, although the relative changes are more sensitive at cooler temperatures. Nonetheless, our findings are overall robust to the choice of reference temperature, for reasonable choices of this parameter over the global ocean. Another simplification is that the effects of salinity changes on the thickness change of a water-mass layer, and thus the heat content change of the layer, have not been separately considered in Eq. (2). Salinity changes can sometimes be large enough to impact the ocean stratification and density structure, and thus influence the ocean circulation and layer thickness[77], impacting the OHC within a given density layer. These salinity-driven impacts on ocean sequestration of heat require further investigation but are beyond the scope of this study.

## Water-mass framework

To study heat content changes within the global water-mass layers and in regional mode and intermediate waters, we first divide the upper 2000 m of the ocean into the tropical water layer (TW, $\gamma^n < 25.0$ kg m$^{-3}$), mode water layer (MW, $25.0 \leq \gamma^n < 27.05$ kg m$^{-3}$), and intermediate water layer (IW, $27.05 \leq \gamma^n < 27.75$ kg m$^{-3}$), as sketched in Fig. 4e and given in Supplementary Tables 1 and 2. The mode water layer is further divided into the Subtropical Mode Water layer (STMW, $25.0 \leq \gamma^n < 26.45$ kg m$^{-3}$) and the Subpolar Mode Water layer (SPMW, $26.45 \leq \gamma^n < 27.05$ kg m$^{-3}$), to separately consider heat content changes in specific Subtropical Mode Waters and Subantarctic Mode Water within these two layers.

The regional water masses of individual mode and intermediate waters are defined within the mode and intermediate water layers, using specific longitude, latitude, and neutral density constraints in Supplementary Table 3 (see also in Figs. 3 and 7). For instance, in the Subtropical Mode Water layer, we separately define regional Subtropical Mode Waters at the subtropical South Atlantic[26,47,78,79] (SA-STMW), North Atlantic[26,47,80–82] (NA-STMW, also known as the North Atlantic's Eighteen Degree Water), western South Pacific[26,47,83,84] (SP-WSTMW), eastern South Pacific[26,47,85] (SP-ESTMW), western North Pacific[26,47,86–88] (NP-STMW), eastern North Pacific[26,47,86,88] (NP-ESTMW), and Indian Ocean[26,47,89,90] (IO-STMW), by further using a set of more specific density and geographic limit (Figs. 3, 7, and 8b, e). The Central Mode Water in the North Pacific[26,47,86,88] (NP-CMW) is also defined within the Subtropical Mode Water layer. In the Subpolar Mode Water layer, we separately define the Subantarctic Mode Water in the Southern Ocean south of 30°S[26,49], and the Madeira Mode Water in the subtropical North Atlantic[26,47,82] (NA-MMW) just beneath the NA-STMW

(Figs. 7 and 8c, e). In the intermediate water layer, we define the Antarctic Intermediate Water in the South Atlantic[26,27,61] due to sustained warming of the intermediate water layer in the Atlantic basin[8] and in the Indian and Pacific sectors of the Southern Ocean (Figs. 7 and 8d, e). We also define the Subpolar Mode Water in the subpolar North Atlantic[26,47,82,91] (NA-SPMW) and the North Pacific Intermediate Water[26,92,93] (NP-IW), which occupies both the Subpolar Mode Water layer and the intermediate water layer. The heat content changes of all these regionally defined modes and intermediate waters are provided in Figs. 8e and 9g, and Supplementary Tables 3 and 4.

Note that the estimate of OHC changes within the mode and intermediate layers and in regionally defined water masses in Figs. 3–9 and Supplementary Tables 1–4 are limited to the Argo era 2005–2020 when the subsurface thermal structure of the ocean was widely observed by the Argo network from 2005. This estimation is based on the ensemble average of SIO RG-Argo, IAP data, and EN4.2.2 ensemble mean. For simplicity, this study uses a set of fixed longitude, latitude, and density or depth criteria to define these water masses and estimate how their OHC has changed over the Argo era, even though the oceanographic features that define them (e.g., temperature, density, salinity and stratification) have also evolved as the upper ocean has warmed and stratified over the Argo era. However, we do nonetheless separate out the volumetric effects from water-mass transformation in attributing total OHC change to each water mass. By decomposing the OHC change into its volumetric and warming components (Eq. 2), the warming component of OHC change within the water-mass layers and in regionally defined water masses (seen in Figs. 5 and 9, and Supplementary Fig. 2 and Tables 2 and 4) is estimated based on a set of fixed longitudes, latitudes, and isopycnal depths (referred to as "fixed volume" or "volume effects removed" in this study) averaged over the Argo era, while the volumetric component is based on a set of density ranges whose depths and volume can evolve over time. This decomposition allows us to distinguish OHC changes of a water mass due to warming effects separate from isopycnal layer thickness (or volume) changes.

## OHC trend and heat uptake

Linear trends of OHC are calculated within each selected time period for spatially integrated annual mean OHC time series, to indicate the increase or decrease in OHC across each selected period. For the trend maps, the fields are first vertically integrated over depth or isopycnal layers, and then the linear trends over the selected period are computed for each grid point. The ocean heat uptake in Fig. 1b is calculated as the linear trend of annual mean OHC for every 11-year period across the measurement era (e.g., 1960–1970, 1970–1980, 1980–1990, 1990–2000, 2000–2010, 2010–2020) multiplied by the length of period. For linear trends within a period of less than 15 years, decadal variability can be part of the signal as well as long-term trends due to anthropogenic warming[17]. Another factor is start- and end-point sensitivity in estimating linear trends and capturing nonlinearity in rates of OHC change[17]. Overall, the monotonic increase in ocean heat uptake has remained clear since the 1980s.

## Data availability

The observational-based data to recreate the figures in this study have been deposited online in the Zenodo database under https://doi.org/10.5281/zenodo.8388661[94]. The gridded SIO RG-Argo product has been continuously updated at http://sio-argo.ucsd.edu/RG_Climatology.html. These data were collected and made freely available by the International Argo Program and the national programs that contribute to it (http://www.argo.ucsd.edu, http://argo.jcommops.org). The Argo Program is part of the Global Ocean Observing System. The global OHC during 1955–2020 from ref. 8 and during 2005–2020 updated by the NOAA/NESDIS/NCEI Ocean Climate Laboratory were obtained from https://www.ncei.noaa.gov/access/

global-ocean-heat-content/heat_global.html. The IAP OHC, temperature and salinity gridded data were obtained from http://www.ocean.iap.ac.cn/. The latest version of EN4 data (EN4.2.2) was downloaded from https://www.metoffice.gov.uk/hadobs/en4/download-en4-2-2.html. The global OHC from ref. 31 was obtained from https://www.data.jma.go.jp/gmd/kaiyou/english/ohc/ohc_global_en.html. The altimeter sea-level anomaly data processed by DUACS and distributed by EU CMEMS can be found at https://marine.copernicus.eu/. TEOS-10 software is available online at http://www.teos-10.org/software.htm. PreTEOS-10 software is available online at http://www.teos-10.org/preteos10_software/neutral_density.html.

## Code availability

All analyses and figures shown in this manuscript are reproducible via MATLAB. Codes needed to process and plot the source data contained in this study are published online in the Zenodo database under https://doi.org/10.5281/zenodo.8388661[94].

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

## Acknowledgements

Z.L. and M.H.E. are supported by the Australian Research Council (SR200100008). M.H.E. also acknowledges past support from the Centre for Southern Hemisphere Oceans Research (CSHOR; a joint research center between QNLM, CSIRO, UNSW and UTAS). Z.L. received support from the Foundation of China Scholarship Council (No. 201806330075). We thank Ivana Cerovečki for helpful discussions on aspects of this work.

## Author contributions

M.H.E. conceived the project. M.H.E., Z.L., and S.G. developed the design of the study and the analyses undertaken. Z.L. undertook the calculations and data analysis, and produced all the graphics used in the study, with input from M.H.E. and S.G. All authors contributed to the analysis, discussion, interpretation and writing of the paper.

## Competing interests

The authors declare no competing interests.
