## [Peer Review File · Nature Communications]

Recent acceleration in global ocean heat accumulation by mode and intermediate watersREVIEWER COMMENTS

Reviewer #1 (Remarks to the Author):

Li et al. investigated the ocean heat content changes in the global mode and intermediate water layers, found that 94% of the total OHC increase are stored in these layers. They further investigated the contribution from thickness change and warming effect, suggesting a ~40% of warming contribution in these layers. The novelty of this study is a comprehensive analysis of OHC change in density layers associated with water masses, which is a valuable perspective and a fresh insight. Thus, I do think the paper is worthy a publication in NC. I have some comments listed below, mainly support to increase the robustness of the quantification and strengthen the conclusion. Some of the comments are about terminology and presentation.

Comments:

- 1) The use of data. This paper mainly relies on Levitus data for long-term changes and Roemmich data for Argo-period analysis. Using single dataset is potential risky because they are suffered from biases, an example is Cheng&Zhu, 2014 GRL study identified a shift in Levitus yearly time series associated with data sampling (2000~2005 during the observation system transfer). It is increasingly clear that sampling is a serious issue. Thus, I would suggest to at least use ensemble mean of several better datasets (Levitus pentadal estimates, Ishii et al. 2017 and Cheng et al. 2017 data). This could provide you more reliable quantification. Roemmich dataset might also not be ideal because of not global coverage (ITF regions, polar regions are not included) and other issues (for instance Trenberth et al. 2016 indicates a quite larger noise level).
- 2) Figure 1, 3 and other places, it is not proper to label 2010-2020 data as Argo because of two reasons: first, Levitus (together with Ishii/Cheng data in my first comment) also used Argo data; second, there are several Argo gridded products, so not only RG09 data is "Argo data". The benefit of Levitus/Ishii/Cheng data is they have real global coverage by merging Argo with other platforms. This could be a thing that impact your quantification. Note that some previous analyses indicate the difference of ocean area could contribute to ~10% of trend difference in OHC.
- 3) Figure 2. It is a bit misleading to add shading area around the time series because it looks like error bars. I suggest to have some rectangle shading in the panel to denote the time periods and add actual error bars if possible.
- 4) Uncertainty assessment should be given for any quantifications in this study. One way is to use the spread among different products.
- 5) "Accelerating global ocean warming" section. Please find a recent study of Cheng et al. 2022 for quantifying the rate change of OHC, this paper could strongly support this part as they do a more thorough analysis to deal with the short-term fluctuations.
- 6) "Levitus period" is defined in this study, I don't support this definition because Levitus does not define any period, it starts from 1955 because of better data coverage.
- 7) Lines 111-117. ENSO: this is not only because of the wind-driven variability, but also air-sea net heat exchanges (see Cheng et al. 2019 ENSO related changes of OHC paper). The thermocline change would not cause net OHC change when averaging over the tropics, the only mechanism can lead to net OHC change is heat exchange at interfaces (e.g. air-sea, tropical vs. middle-latitudes, upper vs deep layers).
- 8) Lines 120-onward. I'm wondering why 2010-2020 period is the focus, the short period means lots of internal variability mixed into anthropogenic signals as you discussed within 133-147.

9) Figure 3 and 4 can be combined to save space: add zonal mean plot to the right side of the spatial maps.

10) Figure 3 vs. Figure 5. Similar to Fig.5, Figure 3 is also a trend map (difference between 2000-2010 and 1980-2000). Comparing the two trend maps indicate quite a significant regional inconsistency, indicating to me that ENSO/IPO could be important at different time periods. Do these variabilities impact your conclusion (i.e. mode/intermediate waters contribution to total OHC)?

11) Table 1. Von Schuckmann et al. 2020 ESSD had a figure nicely summarizing previous efforts and also show the increasing trend of OHC based on a more complete list of literature. I would also remove this table to save space (or move to the SI).

Reviewer #2 (Remarks to the Author):

Review of “Recent acceleration in global ocean heat accumulation by mode and intermediate waters” by Li, England, Groeskamp.

General Comments:

I have enjoyed reading this paper. It has a wide appeal to oceanographers and to a wider communications audience. This appeal is because the manuscript documents in a more thorough way than previously the changes in a key water-mass type that is found widely across the global oceans. The global perspective of this paper means that it details a coherent set of changes in both northern and southern hemispheres. It also shows, quite remarkably in my mind, how this key water mass has actually dominated the total heat uptake in the global oceans, a point that I had known intuitively but had not quite appreciated the scale of that uptake. Finally the heat uptake increase in the most recent period (in both hemispheres, e.g. figure 4 of the manuscript) is also extra-ordinary. The result that these mode waters have an uptake of about 94% of the total ocean heat uptake in these periods from the 0-2000m layer and they occupy just 16% of the global ocean volume. It really means that from the context of ocean heat uptake, these mode (and intermediate) water are the critical water mass.

While not stated, it also implies that the other 84% of the global ocean volume has only received 6% of the net warming. In some ways its also good news, because these mode waters are far from the Poles and therefore “buffering” to some extent the very negative changes that could occur by the melt of the Polar Ice Cap (Antarctica and Greenland), and in terms of the oceanic carbon cycle probably helpful in sequestering anthropogenic carbon in these same regions. Also not stated, but relevant to how we might sample the oceans to monitor it and to “predict” how it might change, thus tracking the earth’s energy imbalance.

So this paper is very interesting for its relevance in the global climate system. It uses relatively standard techniques and approaches to assess the ocean heat content changes for this key (but varied water mass) including the contributions from density (or volume change) and heat storage.

My belief is that this is a very relevant and timely paper for nature communications.

That said, the paper is not without gaps, some minor and some more major (but solvable). This paper is really a statement of the changing state of the upper 2000 metre of the ocean. Its not a full census of the full depth of the ocean. This point should be made more clearly. Unlike modern simulations of the climate system, or re-analysis of the climate system, this paper basically relies on a single gridded data product (RG09) and as support the Satellite Altimetry product (DUACS and distributed by EU CMEMS) and for earlier periods the Levitus product from WODC (reference 5). There are actually at least 6 different gridded products and they do vary between each other when these OHC calculations are done. IPCC would look to provide an ensemble view of the observations rather than relying on a single product. The second gap is that not a single table, or a single graph has any sense of an error bar on them. What is the uncertainty in these 10 year trends for each of the ten year periods provided in this manuscript? Are these trends real at the different layer averages, and different spatial scales, grid point to global averages? Is the heat uptake as extreme as it seems from the abstract and results. Some would say that no errors bars means the authors don't know the uncertainty and so it is very large (rather than very small) and therefore none of the results are valid. My familiarity with these data sets tells me that the results would be replicated across the suite of the products. These two points are part of the solution though. By using the 6 or so available gridded products one can immediately get the ensemble estimate of the changes in these key water masses, and also get from each data set 6 estimates of every quantity presented in the paper and hence the spread and a completely justifiable error bar on every result presented. Perhaps the most important results that would be better for confidence intervals would be Figure 1, Figure 2 panels a,b,c,d, Figure 4, Figure 7 a, Figure 8 g. Table 1, Table 2. The mapped fields/panels in the figures could include the places where change is greater than some confidence interval, but actually I think that is not so important.

Otherwise the figures are quite clear.

While the message of this paper is clear it does leave a number of questions unresolved. It implies that these changes are a simple accelerating trend and are anthropogenically forced. The title says the changes are accelerating. But there is no evidence in this paper that it is an acceleration that can't be explained by say internal variability and climate change (say). The conclusions (line 248 to 254) could be easily rewritten to address whether the increase in uptake relative 1990-2000 is anthropogenically induced or internal variability and climate change. I totally agree its a key water mass in the climate system and the results from this paper are compelling. BUT it could be implausibly high and in 2020-2030 period or the 2030-2040 period we could be writing a similar paper but on the deceleration of mode and intermediate water uptake. I think these are the types of questions this paper should be addressing in lines 248 to 254 (as it sums up the conclusions), and perhaps as a location on the planet that is very important for monitoring the state of the climate.

Specific Comments:

I like the decomposition of Ocean Heat Content in equation 2. It follows earlier work (that is referred to). And in general this equation is spot on, but there have been some papers talking very specifically about the role of ocean heat content in sub-tropical gyres where the salinity changes within the water column are large enough to induce a thickness change and thus on density surface cause cooling signal see on the equatorward side of the gyres in the Indian Ocean and presumably in the Pacific ocean. So there should be a note that the thickness change calculation has not taken into account the effects of salinity changes on thickness change on OHC in the methods section.

Levitus Era or Levitus period (Figure 1 caption). I don't like this terminology. Syd Levitus should have more fame but the period 1955-2010 is not a period of time that we should name after Levitus. We certainly could use the term Levitus Atlas. I would prefer simply "pre Argo", or simply 1955-2010. The IPCC is wary of the ocean thermal data before the 1970's typically because these earlier data are unreliable (see the discussions about xbt and fall rates), and the trends 1960 to 1970 and 1970 to 1980 have clear variability in term. We could also eliminate the "Roemmich period" as well (Figure 2 caption)

Reviewer #3 (Remarks to the Author):

his manuscript analyzes ocean heat content using yearly maps of ocean heat content from 0-2000 m from the World Ocean Atlas 2018 and an updated monthly maps of temperature (as a function of pressure, as well as density) from Roemmich and Gilson (2009). It focuses on contributions of mode and intermediate water warming to ocean warming as a whole. While this work is interesting, and should eventually be suitable for publication, there are substantial issues that preclude publication of the manuscript in its present form. Some of the major issues are expanded upon below, and then a few specific comments are made.

1. First, neither the Roemmich and Gilson (2009) nor the World Ocean Atlas maps are immune to the effects of changes in oceanographic sampling (e.g., Lyman and Johnson, 2014, *J. Climate*). The Roemmich and Gilson (2009) monthly maps relax back to the long-term seasonal climatology in locations and times where data are non-existent or sparse. This means that the first few years of maps are somewhat suspect, as are the global integrals from it in those years as shown in Figures 1 and 2. Similarly, the World Ocean Atlas anomalies in the earlier years are based on very, very sparse data (especially prior to the late 1960s when the XBT came into widespread use), and analyses of these maps should be treated with caution. This means the first decade in figure 1 is very suspect, and really the increasing spatial (and depth of sampling) coverage as time goes on plays a role in the shape of the curve. This should at least be acknowledged in the claims of dramatic acceleration. This is not to say that there is not acceleration, just that some of it may be owing to increases in sampling (both in terms of ocean regions and depths) as the decades go on. It also means that the patterns in the first few years in the panels of Figure 2 are suspect.

2. Second, mode and intermediate waters are defined by oceanographic features that are expected to change density over time. For instance, the temperature, salinity, density, and depth of the salinity minimum that defines the core of the Antarctic Intermediate water has evolved over the decades as the upper ocean has warmed and upper ocean salinity patterns have intensified over the decades (e.g., Schmidtko and Johnson, 2012, *J. Climate*). Similarly, mode waters are defined as a layer of low stratification. The water properties of those mode waters (including their temperature, salinity, depth, or density) will change over time as upper ocean water properties change over time, so their isopycnal boundaries might also change with time. These facts need to be acknowledged, and the interpretation of results (especially in Figure 2f, but also in the maps in Figures 7 and 8) should keep account for this possibility.

3. Third, as the authors are aware, but do not emphasize enough in the manuscript, analyses of ocean heat content in isopycnal layers can be dominated by volume changes between those layers, rather than temperature/salinity changes in those layers. Since the density boundaries defining these mode waters may change with time, one is left asking what these volume changes actually mean? For instance, if the density of the actual mode water shifted across an isopycnal boundary (say it became substantially less dense in a warming world), what would a volume change for that layer mean? Including the volumetric change without changing the isopycnal boundary changes would overemphasize the heat content change.

For the reasons discussed in the preceding two paragraphs it might actually be simpler, and perhaps better, to define a set of fixed depths (they could be defined by isopycnal depths at a specific time, or for a decadal average) to analyse different mode waters, rather than allowing those depths to follow isopycnals as they migrate with time.

5. It is not good practice to name time periods after people. Please find other names than “The Roemmich period” or “The Levitus period”.

6. As presented, the units of J yr^{-1} in Figures 3, 4, 5a, 5c-d, 6, 7, and 8 are all resolution dependent, and need to include area units for the maps (e.g., $\text{J m}^{-2} \text{ yr}^{-1}$) and a length unit for the line plots in Figure 4 ($\text{J m}^{-1} \text{ yr}^{-1}$). Also, the units of Figure 2f are also resolution dependent as presented, and need to be in $\text{J kg}^{-1} \text{ m}^3$.

7. Lines 78-89 and elsewhere: There are a number of different groups estimating ocean warming as is documented in von Shuckman et al. (2020, Earth Syst. Sci. Data), where the acceleration noted here is also discussed in some detail. In addition Loeb et al. (2021, Geophys. Res. Lett.) also document the acceleration of ocean warming 2005-2019, and show that it agrees with increases in net-top-of-the atmosphere energy fluxes observed by satellite.

8. Lines 104-108. Does this perhaps signal actually a change in the density of the mode waters (becoming lighter with time in a warming ocean), rather than a change in volume of intermediate versus mode waters?

9. Line 125. Here it would be appropriate to note the work of Roemmich, who noted previously on that recent warming has been concentrated in the southern Ocean.

10. Lines 170-186. This explication (and others elsewhere which include volume changes) really overemphasizes the importance of warming in the mode and intermediate waters. Volume changes can result in huge “heat” changes, but since heat changes really should be evaluated in a constant volume, they don’t mean much and need to be explained very carefully.

RESPONSE TO REVIEWER COMMENTS:

Below reviewer comments have been copy-pasted in black font, and our responses are indicated in blue font. Line numbers in our responses refer to the revised manuscript (except where indicated).

Reviewer #1 (Remarks to the Author):

Li et al. investigated the ocean heat content changes in the global mode and intermediate water layers, found that 94% of the total OHC increase are stored in these layers. They further investigated the contribution from thickness change and warming effect, suggesting a ~40% of warming contribution in these layers. The novelty of this study is a comprehensive analysis of OHC change in density layers associated with water masses, which is a valuable perspective and a fresh insight. Thus, I do think the paper is worthy a publication in NC. I have some comments listed below, mainly support to increase the robustness of the quantification and strengthen the conclusion. Some of the comments are about terminology and presentation.

Response: We thank Reviewer #1 for the positive assessment and detailed comments in this round of review. We believe these comments have significantly helped us to improve the paper.

Reviewer comments:

1) The use of data. This paper mainly relies on Levitus data for long-term changes and Roemmich data for Argo-period analysis. Using single dataset is potential risky because they are suffered from biases, an example is Cheng&Zhu, 2014 GRL study identified a shift in Levitus yearly time series associated with data sampling (2000~2005 during the observation system transfer). It is increasingly clear that sampling is a serious issue. Thus, I would suggest to at least use ensemble mean of several better datasets (Levitus pentadal estimates, Ishii et al. 2017 and Cheng et al. 2017 data). This could provide you more reliable quantification. Roemmich dataset might also not be ideal because of not

global coverage (ITF regions, polar regions are not included) and other issues (for instance Trenberth et al. 2016 indicates a quite larger noise level).

Response: Thanks for pointing this out and providing the insightful suggestions. This comment picks up on points also raised by Reviewer 2. We agree that this is a possible source of uncertainty by relying only on Levitus data for the pre-Argo period analysis (1955–2004) and Roemmich Argo data for the Argo-period analysis (2005–2020). To address this issue and improve the robustness of our results, we have now included observational-based ocean heat content (OHC) products from Levitus et al. (2012), Cheng et al. (2017), and Ishii et al. (2017), and also included the estimate of global and regional OHC using temperature and salinity observations from SIO RG-Argo, IAP data (Cheng and Zhu 2016; Cheng et al. 2017), and the ensemble mean of Met Office Hadley Centre EN4.2.2 data (Good et al., 2013), into the manuscript. We have now added a detailed description of this point in *Methods* of the revised manuscript (lines 356–404) and updated all results and display diagrams in the manuscript accordingly.

We have also included a statement of the uncertainty that may come from using SIO RG-Argo, which does not have a global coverage, on lines 367–371: “We only use the SIO RG-Argo data between 65°S–65°N because of the low observational data coverage in high latitude regions, particularly ice-covered areas and in marginal seas. As such, the resultant global OHC trends from SIO RG-Argo in Fig. 1 will be underestimated if these polar ocean regions are warming⁵.” Also, a clarification of the shift in OHC estimates that is due to the change in locations of the observational network has been added to lines 128–131: “For example, increased spatial sampling coverage, particularly over the tropical and Southern Hemisphere oceans due to the shift in location of the observational network from a ship-based system to the Argo network, can affect the estimate of ocean warming during 2001–2003³⁴ (Fig. 1, Supplementary Fig. 1).”.

2) Figure 1, 3 and other places, it is not proper to label 2010–2020 data as Argo because of two reasons: first, Levitus (together with Ishii/Cheng data in my first

comment) also used Argo data; second, there are several Argo gridded products, so not only RG09 data is “Argo data”. The benefit of Levitus/Ishii/Cheng data is they have real global coverage by merging Argo with other platforms. This could be a thing that impact your quantification. Note that some previous analyses indicate the difference of ocean area could contribute to ~10% of trend difference in OHC.

Response: Thanks for the suggestions. We have now referred to the gridded Argo product from Roemmich and Gilson (2009) as Scripps Institution of Oceanography (SIO) RG-Argo, and have additionally included the EN4.2.2 ensemble mean from the Met Office Hadley Center (Good et al., 2013), the IAP data (Cheng and Zhu 2016; Cheng et al. 2017), and the Ishii et al. (2017) data into our analyses. These datasets are constructed by merging both Argo and other historical observations and are all global in coverage. As such, the inclusion of these datasets provides a more robust estimate of global OHC changes. We have clarified this point on lines 367–371 “This gridded Argo product was constructed from Argo data only. We only use the SIO RG-Argo data between 65°S-65°N because of the low observational data coverage in high latitude regions, particularly ice-covered areas and in marginal seas. As such, the resultant global OHC trends from SIO RG-Argo in Fig. 1 will be underestimated if these polar ocean regions are warming⁵.”

It may also be noted that in Figs. 2–10 of the manuscript, where we show OHC changes in global mode and intermediate layers and within localized mode and intermediate waters, we focus on the region between 65°S-65°N due to the low sampling coverage at higher latitudes and for better consistency of the data’s spatial coverage, as now clarified in the caption of Figs. 2, 3 and Supplementary Table 1, and on line 135–138: “Therefore, this study mainly focuses on the Argo era 2005–2020, an overlapping period common to all products analyzed, and on the ocean domain between 65°S and 65°N, because of the sparse observations in the polar regions even after the Argo array was established.”.

3) Figure 2. It is a bit misleading to add shading area around the time series because it

looks like error bars. I suggest to have some rectangle shading in the panel to denote the time periods and add actual error bars if possible.

Response: We thank the reviewer for pointing this out. We have now included the OHC estimate based on temperature and salinity observations from SIO RG-Argo, the IAP data (Cheng and Zhu 2016; Cheng et al. 2017), and EN4.2.2 ensemble mean from the Met Office Hadley Center (Good et al., 2013), into the Argo period analyses, and have updated Figs. 2–5, 9, 10, Supplementary Fig. 1, and Supplementary Tables 1–4, regarding this point (see more details in *Methods*, lines 356–404). Now both the OHC estimate from each of these three products and their ensemble mean are provided in Fig. 2. The uncertainty range (± 2 standard deviations) from these three products is provided in Figs. 3a-d and Supplementary Tables 1, 2.

4) Uncertainty assessment should be given for any quantifications in this study. One way is to use the spread among different products.

Response: Thanks for the suggestion. We have now included observational-based OHC products from Levitus et al. (2012), Cheng et al. (2017), and Ishii et al. (2017), and also included the estimate of global and regional OHC using temperature and salinity observations from SIO RG-Argo, IAP data (Cheng and Zhu 2016; Cheng et al. 2017), and the ensemble mean of Met Office Hadley Centre EN4.2.2 data (Good et al., 2013), into the manuscript. We have now updated all results and figures in the manuscript based on the inclusion of these new datasets. Now both the ensemble mean and the uncertainty range ($\pm 2\sigma$) from all products are provided in the manuscript. We have now included a statement of this point on lines 101–104: “To provide a more reliable analysis of the global and regional OHC changes in this study, we present the ensemble average of OHC estimates derived from various ocean data products (Fig. 1; Methods). The uncertainty range is given by ± 2 standard deviations ($\pm 2\sigma$) to indicate the degree of agreement across the products.”, in figure and table captions, and on lines 397–404.

5) “Accelerating global ocean warming” section. Please find a recent study of Cheng et

al. 2022 for quantifying the rate change of OHC, this paper could strongly support this part as they do a more thorough analysis to deal with the short-term fluctuations.

Response: Thanks for the suggestions. We have now added a citation to Cheng et al. 2022 (*Journal of Climate*) in the manuscript (e.g., lines 50, 107) and also added a clarification of this point in this section (lines 114–117): “Further comparison with previous estimates of OHC trends (Table 1) and the latest assessment report of the Intergovernmental Panel on Climate Change (IPCC) Working Group I⁷ also reveals significant increases in the ocean warming rate across the measurement record^{5,8–11,16,17,19.}”, and on lines 126–127: “Another factor is start- and end-point sensitivity in estimating linear trends and capturing nonlinearity in rates of OHC change^{16.}”

6) “Levitus period” is defined in this study, I don’t support this definition because Levitus does not define any period, it starts from 1955 because of better data coverage.

Response: Agreed – thanks, good suggestion. We have now referred to the period 1955–2004 as the ‘pre-Argo period’ and the period 2005–2020 as the ‘Argo era’ in the manuscript. The terminologies of ‘Levitus period’ and ‘Roemmich period’ have all been removed from the manuscript.

7) Lines 111-117. ENSO: this is not only because of the wind-driven variability, but also air-sea net heat exchanges (see Cheng et al. 2019 ENSO related changes of OHC paper). The thermocline change would not cause net OHC change when averaging over the tropics, the only mechanism can lead to net OHC change is heat exchange at interfaces (e.g. air-sea, tropical vs. middle-latitudes, upper vs deep layers).

Response: Agreed - thanks for raising this. We have now rephrased the description of ENSO related changes of OHC in this section (lines 162–164 of revised manuscript): “This ENSO-driven variability in OHC is related to net air-sea heat exchanges and the redistribution of heat laterally across different latitudes and ocean regions, and vertically between surface and subsurface layers^{35,36.}”, and added a citation to Cheng et al. (2019) in the manuscript (lines 162, 164).

8) Lines 120-onward. I'm wondering why 2010-2020 period is the focus, the short period means lots of internal variability mixed into anthropogenic signals as you discussed within 133-147.

Response: Agreed, part of this came about because we initially focused on the decade-by-decade accumulation of heat in the ocean, but the downside is a signal of internal variability, as the reviewer notes. To reduce the impacts from internal variability and take advantage of available observations, we have now changed our analysis period to focus on the Argo era 2005–2020 (an overlapping period of all datasets used), when Argo data became mature in spatial coverage from 2005. We have now updated Figs. 2–4, 6–10 and Supplementary Tables 1–4 accordingly. We have now added a statement in regard to this point on lines 131–138: “Even though all products show robust ocean warming rates for the upper 2000 m over the past few decades, reaching record warming over 2010–2020^{5,16} (Table 1), these products reach better agreement after ~2005 when the Argo network was widely deployed. To minimize the impact of internal climate variability and focus on longer-term trends, we consider the longest reliable OHC record available. Therefore, this study mainly focuses on the Argo era 2005–2020, an overlapping period common to all products analyzed, and on the ocean domain between 65°S and 65°N, because of the sparse observations in the polar regions even after the Argo array was established.”.

9) Figure 3 and 4 can be combined to save space: add zonal mean plot to the right side of the spatial maps.

Response: Agreed - good suggestion, thanks. We have now combined the previous Figs. 3 and 4 as the new Fig. 5. Due to the inclusion of additional observational-based datasets, the ensemble average of decadal mean OHC change is now shown in the spatial maps (Figs. 5a-b), and both the ensemble mean and the uncertainty range ($\pm 2\sigma$) are provided in the zonal mean plot (Figs. 5c-d). This is a nice improvement to these analyses.

10) Figure 3 vs. Figure 5. Similar to Fig.5, Figure 3 is also a trend map (difference between 2000-2010 and 1980-2000). Comparing the two trend maps indicate quite a significant regional inconsistency, indicating to me that ENSO/IPO could be important at different time periods. Do these variabilities impact your conclusion (i.e. mode/intermediate waters contribution to total OHC)?

Response: Thanks for raising this. We agree that there are regional inconsistencies in OHC changes between Fig. 3 (now Fig. 5) and Fig. 5 (now Fig. 6). This occurs mainly over the tropical Pacific Ocean, whereas the rest of the Pacific Ocean and the Indian, Atlantic, and Southern Oceans tend to show a consistent and robust warming, in particular over subduction/ventilation regions near the western boundary current extensions and the ACC (Figs. 5a-b, 6a, 8; Rathore et al. 2020; Zika et al. 2020). This robust long-term warming over the subtropical Atlantic and Pacific Oceans and in the Southern Ocean, as well as the near-monotonic warming in the mode and intermediate water layers across the Argo era (Figs. 2, 3a; line 150–153), suggest a robust ocean heat gain by mode and intermediate waters. In contrast, OHC changes within the tropical water layer and in the tropical Pacific Ocean reveal large variations associated with ENSO and/or IPO variability (Figs. 2b, 3b, 5a-b, 6a; lines 158–169 and 187–194).

We further agree that the formation of mode and intermediate waters can be influenced by interannual to decadal variability of the climate system. For example, interannual variations in the formation of South Pacific SAMW have been linked to tropical teleconnections from the Indo-Pacific (Li and England 2020). In order to reduce the footprint from climate modes to our conclusion, we have now changed our main water-mass analyses to focus on a longer period over the Argo era 2005–2020 (an overlapping period of all the datasets analyzed). We have now added a clarification of this point on lines 131–138 and lines 294–307: “While the globally integrated OHC of the upper 2000 m and in the mode and intermediate water layers both show a robust and near-monotonic increase (Figs. 1–3), variations in the formation mechanisms, properties, and heat content of localized mode and intermediate waters can be influenced by interannual to decadal variability (e.g. ENSO and the Indian Ocean

Dipole⁶⁰ on interannual time-scales, and the IPO on decadal time-scales). However, our analyses show that the main footprint of internal climate variability occurs over the tropical Pacific Ocean, whereas the rest of the Pacific as well as the Indian, Atlantic, and Southern Oceans all tend to show a consistent and robust warming, in particular over the subduction and well-ventilated regions near western boundary current extensions and north of the ACC (compare Figs. 5a-b and 6a). The near-monotonic warming in mode and intermediate water layers across the Argo era (Figs. 2, 3a), suggests a robust ocean heat gain forced by anthropogenic change. In contrast, OHC changes within the tropical water layer and in the tropical Pacific Ocean reveal variations likely associated with ENSO and/or IPO variability.”. Text has also been added on this point at lines 328–331: “Although our results are robust within a set of observational-based datasets, important questions remain, such as separating out the anthropogenic signal of OHC change from internal variability, particularly in mode and intermediate waters where decadal variability is poorly constrained by sparse measurements prior to the Argo era.”

We have also added text on the point of continuing establishing and improving subtropical, polar, and deep ocean observing systems, which is of fundamental importance for better monitoring the state of the ocean and understanding the mechanisms that are driving ongoing ocean warming, at lines 325–342 and 343–347, as per the suggestion from Reviewer 2.

11) Table 1. Von Schuckmann et al. 2020 ESSD had a figure nicely summarizing previous efforts and also show the increasing trend of OHC based on a more complete list of literature. I would also remove this table to save space (or move to the SI).

Response: Thanks for the suggestion. We have now updated Table 1 and decided to keep it in the main manuscript for readers to have ready access to this table showing accelerating ocean warming built on past studies. So we feel it’s worth keeping the Table. We have nonetheless now included a citation to von Shuckman et al. (2020) and the latest assessment report of the IPCC Working Group I in the manuscript text, lines 114–123: “Further comparison with previous estimates of OHC trends (Table 1) and the

latest assessment report of the Intergovernmental Panel on Climate Change (IPCC) Working Group I⁷ also reveals significant increases in the ocean warming rate across the measurement record^{5,8–11,16,17,19}. This increasing ocean warming rate is not just limited to the upper layer of the ocean, but also occurs at intermediate depths (700–2000 m) according to a further breakdown of OHC trends by depth over the upper 2000 m^{5,7}. Significant ocean warming and accelerating OHC changes are consistent with the increase in net radiative energy absorbed by Earth detected in satellite observations^{5,33}, something that is likely to continue throughout the 21st Century^{6,15} in the absence of substantial greenhouse gas emissions reductions.”.

Reviewer #2 (Remarks to the Author):

Review of “Recent acceleration in global ocean heat accumulation by mode and intermediate waters” by Li, England, Groeskamp.

General Comments:

I have enjoyed reading this paper. It has a wide appeal to oceanographers and to a wider communications audience. This appeal is because the manuscript documents in a more thorough way than previously the changes in a key water-mass type that is found widely across the global oceans. The global perspective of this paper means that it details a coherent set of changes in both northern and southern hemispheres. It also shows, quite remarkably in my mind, how this key water mass has actually dominated the total heat uptake in the global oceans, a point that I had known intuitively but had not quite appreciated the scale of that uptake. Finally the heat uptake increase in the most recent period (in both hemispheres, e.g. figure 4 of the manuscript) is also extraordinary. The result that these mode waters have an uptake of about 94% of the total ocean heat uptake in these periods from the 0-2000m layer and they occupy just 16% of the global ocean volume. It really means that from the context of ocean heat uptake, these mode (and intermediate) water are the critical water mass.

While not stated, it also implies that the other 84% of the global ocean volume has only received 6% of the net warming. In some ways its also good news, because these mode waters are far from the Poles and therefore “buffering” to some extent the very negative changes that could occur by the melt of the Polar Ice Cap (Antarctica and Greenland), and in terms of the oceanic carbon cycle probably helpful in sequestering anthropogenic carbon in these same regions. Also not stated, but relevant to how we might sample the oceans to monitor it and to “predict” how it might change, thus tracking the earth’s energy imbalance.

So this paper is very interesting for its relevance in the global climate system. It uses relatively standard techniques and approaches to assess the ocean heat content changes for this key (but varied water mass) including the contributions from density (or

volume change) and heat storage.

My belief is that this is a very relevant and timely paper for nature communications.

Response: We thank Reviewer #2 for this favourable assessment of our paper, and for their insightful comments in this round of review. We believe these comments have helped us to improve the paper. Thank you.

That said, the paper is not without gaps, some minor and some more major (but solvable). This paper is really a statement of the changing state of the upper 2000 metre of the ocean. Its not a full census of the full depth of the ocean. This point should be made more clearly.

Response: We thank the reviewer for pointing this out. We have now added text to clarify that the focus of this study is the changing state of the upper 2000 m of the ocean in Methods (lines 412–418): “This study focuses on warming of the upper 2000 m of the ocean in part because this is the volume of the ocean in closest contact with the atmosphere, to experience global warming. This 2000-m constraint is also due to the depth limit of the Argo profiling floats, with only sparse data coverage below 2000 m. Waters below 2000 m, apart from recently overturned Antarctic Bottom Water and Lower North Atlantic Deep Water, are generally not ventilated rapidly enough to have experienced the last 50 years of global warming.”.

Unlike modern simulations of the climate system, or re-analysis of the climate system, this paper basically relies on a single gridded data product (RG09) and as support the Satellite Altimetry product (DUACS and distributed by EU CMEMS) and for earlier periods the Levitus product from WODC (reference 5). There are actually at least 6 different gridded products and they do vary between each other when these OHC calculations are done. IPCC would look to provide an ensemble view of the observations rather than relying on a single product. The second gap is that not a single table, or a single graph has any sense of an error bar on them. What is the uncertainty

in these 10 year trends for each of the ten year periods provided in this manuscript? Are these trends real at the different layer averages, and different spatial scales, grid point to global averages? Is the heat uptake as extreme as it seems from the abstract and results. Some would say that no errors bars means the authors don't know the uncertainty and so it is very large (rather than very small) and therefore none of the results are valid. My familiarity with these data sets tells me that the results would be replicated across the suite of the products. These two points are part of the solution though. By using the 6 or so available gridded products one can immediately get the ensemble estimate of the changes in these key water masses, and also get from each data set 6 estimates of every quantity presented in the paper and hence the spread and a completely justifiable error bar on every result presented. Perhaps the most important results that would be better for confidence intervals would be Figure 1, Figure 2 panels a,b,c,d, Figure 4, Figure 7 a, Figure 8 g. Table 1, Table 2. The mapped fields/panels in the figures could include the places where change is greater than some confidence interval, but actually I think that is not so important.

Otherwise the figures are quite clear.

Response: Thanks for pointing this out and providing these useful suggestions. This comment picks up on points also raised by Reviewer 1. We agree that this is a possible source of uncertainty by relying only on Levitus data for the pre-Argo period analysis (1955–2004) and Roemmich Argo data for the Argo-period analysis (2005–2020). To address this issue and improve the robustness of our results, we have now included observational-based ocean heat content (OHC) products from Levitus et al. (2012), Cheng et al. (2017), and Ishii et al. (2017), and also included the estimate of global and regional OHC using temperature and salinity observations from SIO RG-Argo, IAP data (Cheng and Zhu 2016; Cheng et al. 2017), and the ensemble mean of Met Office Hadley Centre EN4.2.2 data (Good et al., 2013), into the manuscript. We have now added a detailed description of this point in *Methods* of the revised manuscript (lines 356–404) and updated all results and display diagrams in the manuscript accordingly.

In addition, the ensemble mean estimate of global OHC changes and OHC changes in key water masses from these products is now provided throughout the manuscript, instead of the previous approach, and the uncertainty range ($\pm 2\sigma$) is provided in Figs. 1, 3a-d, 5c-d, 9a, 10g, and S1 and Tables 1, S1–4.

While the message of this paper is clear it does leave a number of questions unresolved. It implies that these changes are a simple accelerating trend and are anthropogenically forced. The title says the changes are accelerating. But there is no evidence in this paper that it is an acceleration that can't be explained by say internal variability and climate change (say). The conclusions (line 248 to 254) could be easily rewritten to address whether the increase in uptake relative 1990-2000 is anthropogenically induced or internal variability and climate change. I totally agree its a key water mass in the climate system and the results from this paper are compelling. BUT it could be implausibly high and in 2020-2030 period or the 2030-2040 period we could be writing a similar paper but on the deceleration of mode and intermediate water uptake. I think these are the types of questions this paper should be addressing in lines 248 to 254 (as it sums up the conclusions), and perhaps as a location on the planet that is very important for monitoring the state of the climate.

Response: We agree that OHC trends within this relatively short time period can indicate a combination of both an anthropogenic forced warming signal as well as a signal due to internal variability (lines 187–188). For example, the OHC changes within the tropical water layer and in the tropical Pacific Ocean reveal large variations associated with ENSO and/or IPO variability (Figs. 2b, 3b, 5a-b, 6a; lines 158–167 and 187–194), while the mode and intermediate water layers exhibit less variability and more steady warming across the Argo era (see Figs. 2, 3a; line 150–153), especially for warming of the intermediate water layer (Figs. 3d, e). Overall, the subtropical Pacific, Indian, and Atlantic Oceans, and the Southern Ocean, all show a consistent and robust warming, in particular over subduction/ventilation regions near the western boundary current extensions and the ACC (Figs. 5a-b, 6a, 8; Rathore et al. 2020; Zika et al. 2020). This robust long-term warming over the subtropical Atlantic and Pacific Oceans

and in the Southern Ocean, as well as the near-monotonic warming in the mode and intermediate water layers across the Argo era (Figs. 2, 3a; line 150–153), suggest a robust ocean heat gain by mode and intermediate waters.

We further agree that the formation of mode and intermediate waters can be influenced by interannual to decadal variability of the climate system. For example, interannual variations in the formation of South Pacific SAMW have been linked to tropical teleconnections from the Indo-Pacific (Li and England 2020). In order to reduce the footprint from climate modes to our conclusion, we have now changed our main water-mass analyses to focus on a longer period over the Argo era 2005–2020 (an overlapping period of all the datasets analyzed). We have now added a clarification of this point on lines 131–138 and lines 294–307: “While the globally integrated OHC of the upper 2000 m and in the mode and intermediate water layers both show a robust and near-monotonic increase (Figs. 1–3), variations in the formation mechanisms, properties, and heat content of localized mode and intermediate waters can be influenced by interannual to decadal variability (e.g. ENSO and the Indian Ocean Dipole⁶⁰ on interannual time-scales, and the IPO on decadal time-scales). However, our analyses show that the main footprint of internal climate variability occurs over the tropical Pacific Ocean, whereas the rest of the Pacific as well as the Indian, Atlantic, and Southern Oceans all tend to show a consistent and robust warming, in particular over the subduction and well-ventilated regions near western boundary current extensions and north of the ACC (compare Figs. 5a-b and 6a). The near-monotonic warming in mode and intermediate water layers across the Argo era (Figs. 2, 3a), suggests a robust ocean heat gain forced by anthropogenic change. In contrast, OHC changes within the tropical water layer and in the tropical Pacific Ocean reveal variations likely associated with ENSO and/or IPO variability.”.

Text on this point has also been added at lines 343–347: “The world’s oceans hold by far the largest reservoir of excess anthropogenic heat in the Earth system¹. Knowing exactly where and how much heat is stored in the ocean is thus fundamental to understanding how climate and sea-level rise will change in the future. This knowledge is also vital to designing the optimal observing networks for monitoring the state of the

ocean and climate system.”, and lines 325–342: “Our analysis of heat accumulation within these key water masses focused on the Argo era 2005–2020, when the subsurface thermal structure of the ocean is well-constrained by the Argo network. Although our results are robust within a set of observational-based datasets, important questions remain, such as separating out the anthropogenic signal of OHC change from internal variability, particularly in mode and intermediate waters where decadal variability is poorly constrained by sparse measurements prior to the Argo era. In addition, due to sparse data coverage over the ice-covered oceans, in marginal seas, in the deep and abyssal oceans, and in the years prior to the Argo network, not all regions of the ocean or time-scales are well-constrained by a suitably dense array of measurements. It is thus essential to continue establishing and improving coastal, polar, and deep ocean observing systems to better monitor changes of the ocean and track the global energy balance more reliably. Nonetheless, our work reveals accelerating warming in the upper 2000 m of the ocean over the past several decades and highlights the increasing heat uptake by mode and intermediate waters, as well as the drivers regulating the formation of, and heat uptake in, these water masses. Regional observations of subsurface temperature and salinity in the subtropical and subpolar oceans are thus of fundamental importance to better understand the mechanisms that are driving ongoing warming and predict the state of climate in the future.”.

Specific Comments:

I like the decomposition of Ocean Heat Content in equation 2. It follows earlier work (that is referred to). And in general this equation is spot on, but there have been some papers talking very specifically about the role of ocean heat content in sub-tropical gyres where the salinity changes within the water column are large enough to induce a thickness change and thus on density surface cause cooling signal see on the equatorward side of the gyres in the Indian Ocean and presumably in the Pacific Ocean. So there should be a note that the thickness change calculation has not taken into account the effects of salinity changes on thickness change on OHC in the methods section.

Response: Agreed – thanks for the suggestions. We have now added a statement of this point on lines 449–455: “Another simplification is that the effects of salinity changes on the thickness change of a water-mass layer, and thus the heat content change of the layer, have not been separately considered in Eq. (2). Salinity changes can sometimes be large enough to impact the ocean stratification and density structure, and thus influence the ocean circulation and layer thickness⁶⁸, impacting the OHC within a given density layer. These salinity-driven impacts on ocean sequestration of heat require further investigation, but are beyond the scope of this study.”.

Levitus Era or Levitus period (Figure 1 caption). I don't like this terminology. Syd Levitus should have more fame but the period 1955-2010 is not a period of time that we should name after Levitus. We certainly could use the term Levitus Atlas. I would prefer simply “pre Argo”, or simply 1955-2010. The IPCC is wary of the ocean thermal data before the 1970's typically because these earlier data are unreliable (see the discussions about xbt and fall rates), and the trends 1960 to 1970 and 1970 to 1980 have clear variability in term. We could also eliminate the “Roemmich period” as well (Figure 2 caption)

Response: Agreed – thanks for pointing this out. We have now updated the terminology for the period 1955-2004 as “pre-Argo period” and the period 2005-2020 as “Argo era” in the manuscript. The terminologies of “Levitus period” and “Roemmich period” in the previous version of the manuscript have now all been removed.

Reviewer #3 (Remarks to the Author):

This manuscript analyzes ocean heat content using yearly maps of ocean heat content from 0-2000 m from the World Ocean Atlas 2018 and an updated monthly maps of temperature (as a function of pressure, as well as density) from Roemmich and Gilson (2009). It focuses on contributions of mode and intermediate water warming to ocean warming as a whole. While this work is interesting, and should eventually be suitable for publication, there are substantial issues that preclude publication of the manuscript in its present form. Some of the major issues are expanded upon below, and then a few specific comments are made.

Response: We thank Reviewer #3 for the insightful and detailed comments that helped us to significantly improve the paper.

1. First, neither the Roemmich and Gilson (2009) nor the World Ocean Atlas maps are immune to the effects of changes in oceanographic sampling (e.g., Lyman and Johnson, 2014, J. Climate). The Roemmich and Gilson (2009) monthly maps relax back to the long-term seasonal climatology in locations and times where data are non-existent or sparse. This means that the first few years of maps are somewhat suspect, as are the global integrals from it in those years as shown in Figures 1 and 2. Similarly, the World Ocean Atlas anomalies in the earlier years are based on very, very sparse data (especially prior to the late 1960s when the XBT came into widespread use), and analyses of these maps should be treated with caution. This means the first decade in figure 1 is very suspect, and really the increasing spatial (and depth of sampling) coverage as time goes on plays a role in the shape of the curve. This should at least be acknowledged in the claims of dramatic acceleration. This is not to say that there is not acceleration, just that some of it may be owing to increases in sampling (both in terms of ocean regions and depths) as the decades go on. It also means that the patterns in the first few years in the panels of Figure 2 are suspect.

Response: Thanks for pointing this out. We have now added a clarification of this point on lines 95–107: “The community has also made considerable progress in developing

gridded products of subsurface temperature for the global ocean by merging all observing systems^{8,10,30,31}. Due to instrumental biases in bathythermograph measurements³² and insufficient data coverage prior to Argo (before 2005), as well as geographic and depth limitations of conventional Argo floats, various correction schemes and reanalysis methods have been applied to mitigate these biases and increase our confidence in the OHC estimate. To provide a more reliable analysis of the global and regional OHC changes in this study, we present the ensemble average of OHC estimates derived from various ocean data products (Fig. 1; Methods). The uncertainty range is given by ± 2 standard deviations ($\pm 2\sigma$) to indicate the degree of agreement across the products. Although some discrepancies of the global OHC estimates are evident among the different datasets, the ensemble mean of OHC estimates for the upper 2000 m shows that ocean warming is not only continuing, but accelerating^{5,8,16} (Fig. 1).”, and lines 124–134: “While this is a striking acceleration of global ocean warming that warrants further investigation, uncertainties due to data scarcity and data quality prior to the late 1960s¹⁰ should be factored in when interpreting these results. Another factor is start- and end-point sensitivity in estimating linear trends and capturing nonlinearity in rates of OHC change¹⁶. For example, increased spatial sampling coverage, particularly over the tropical and Southern Hemisphere oceans due to the shift in location of the observational network from a ship-based system to the Argo network, can affect the estimate of ocean warming during 2001–2003³⁴ (Fig. 1, Supplementary Fig. 1). Even though all products show robust ocean warming rates for the upper 2000 m over the past few decades, reaching record warming over 2010–2020^{5,16} (Table 1), these products reach better agreement after ~2005 when the Argo network was widely deployed.”

To reduce the uncertainty from only using SIO RG-Argo and Levitus data, and improve the robustness of our results, we have now additionally included observational-based OHC products from Cheng et al. (2017) and Ishii et al. (2017), and also included the estimate of global and regional OHC using temperature and salinity observations from IAP data (Cheng and Zhu 2016; Cheng et al. 2017), and the ensemble mean of Met Office Hadley Centre EN4.2.2 data (Good et al., 2013), into the manuscript. These datasets are constructed from Argo and other historical observations, and are based on

various correction approaches by the community to correct biases in XBT and MBT data. We have added a detailed description of this point in *Methods* (lines 356–404) and updated all results and figures in the revised manuscript accordingly. Now the ensemble mean estimate of global OHC changes and OHC changes in key water masses from these products is provided throughout the manuscript, and the uncertainty range ($\pm 2\sigma$) is provided in Figs. 1, 3a-d, 5c-d, 9a, 10g, S1 and Tables 1, S1–4.

2. Second, mode and intermediate waters are defined by oceanographic features that are expected to change density over time. For instance, the temperature, salinity, density, and depth of the salinity minimum that defines the core of the Antarctic Intermediate water has evolved over the decades as the upper ocean has warmed and upper ocean salinity patterns have intensified over the decades (e.g., Schmidtko and Johnson, 2012, *J. Climate*). Similarly, mode waters are defined as a layer of low stratification. The water properties of those mode waters (including their temperature, salinity, depth, or density) will change over time as upper ocean water properties change over time, so their isopycnal boundaries might also change with time. These facts need to be acknowledged, and the interpretation of results (especially in Figure 2f, but also in the maps in Figures 7 and 8) should keep account for this possibility.

Response: Thanks for the suggestions. We agree that the properties of mode and intermediate waters (including their temperature, salinity, depth, PV and stratification) have changed as the upper ocean has warmed and stratified over the past decades. So we have now included clarifications of this point on lines 319–325: “Mode and intermediate waters are traditionally defined by their oceanographic features such as temperature, density, salinity, and stratification. These features have evolved as the upper ocean has warmed and stratified over the last few decades, and as precipitation minus evaporation fields have changed. Here for simplicity, we use a set of fixed longitude, latitude, and density or depth criteria to define these water masses and estimate how their OHC has changed over the Argo era, although we do separate out the effects of volumetric changes from water-mass transformation in attributing total OHC change to each water mass.”, and in *Methods* (lines 491–497): “For simplicity this

study uses a set of fixed longitude, latitude, and density or depth criteria to define these water masses and estimate how their OHC has changed over the Argo era, even though the oceanographic features that define them (e.g. temperature, density, salinity and stratification) have also evolved as the upper ocean has warmed and stratified over the Argo era. However, we do nonetheless separate out the volumetric effects from water-mass transformation in attributing total OHC change to each water mass.”.

3. Third, as the authors are aware, but do not emphasize enough in the manuscript, analyses of ocean heat content in isopycnal layers can be dominated by volume changes between those layers, rather than temperature/salinity changes in those layers. Since the density boundaries defining these mode waters may change with time, one is left asking what these volume changes actually mean? For instance, if the density of the actual mode water shifted across an isopycnal boundary (say it became substantially less dense in a warming world), what would a volume change for that layer mean? Including the volumetric change without changing the isopycnal boundary changes would overemphasize the heat content change.

Response: Thanks for pointing this out. We agree that OHC changes of a mode water density layer can often be dominated by its volumetric changes. Volume expansion of mode waters suggests more oceanic heat and carbon has been injected into the ocean interior, while warming is of fundamental importance as the subducted mode water can warm the surrounding water masses in the ocean interior via mixing (lines 261–264). By decomposing the OHC change into its volumetric and warming components (Eq. 2), we have decomposed OHC changes due to volumetric changes of the density layer (e.g., blue dashed lines in Fig. 3) separately from net warming of the layer (calculated using fixed volume based on the Argo era average density field; e.g., red dashed lines in Fig. 3; Figs. 4, 10, and Tables S2, S4). We have now added further clarification of this point on lines 433–438 and lines 497–505 of the revised manuscript: “By decomposing the OHC change into its volumetric and warming components (Eq. 2), the warming component of OHC change within the water-mass layers and in regionally defined water masses (seen in Figs. 3, 4, 10, and Supplementary Tables 2, 4) is estimated based on a

set of fixed longitudes, latitudes, and isopycnal depths (referred to as “fixed volume” or “volume effects removed” in this study) averaged over the Argo era, while the volumetric component is based on a set of density ranges whose depths and volume can evolve over time. This decomposition allows us to distinguish OHC changes of a water mass due to warming effects separate from isopycnal layer thickness (or volume) changes.”. Then we have rephrased statements in the manuscript to highlight the role of net warming (fixed volume) in the OHC change and to avoid overemphasizing the heat content change in mode and intermediate waters, such as on lines 150–157 and 226–240.

4. For the reasons discussed in the preceding two paragraphs it might actually be simpler, and perhaps better, to define a set of fixed depths (they could be defined by isopycnal depths at a specific time, or for a decadal average) to analyse different mode waters, rather than allowing those depths to follow isopycnals as they migrate with time.

Response: Thanks for the suggestions. Actually by decomposing the OHC change into its volumetric and warming components (Eq. 2), the warming component of OHC change within the water-mass layers and in localized water masses (Figs. 3–4, 10, and Tables S2, S4) can be estimated based on a set of fixed longitudes, latitudes, and isopycnal depths averaged over the Argo era, while the volumetric component is based on a set of density ranges whose depth and volume can migrate with time. This decomposition of the OHC change into its warming component at fixed isopycnal depths and volumetric component at fixed temperatures provides a more complete analysis of OHC changes in water masses. We have now added a statement of this point at lines 433–438: “The terms \bar{z}_1 and \bar{z}_2 indicate the fixed location and depth/thickness of a density layer over the Argo era, based on the climatologically averaged density field. Thus the warming component of OHC changes is estimated based on fixed (Argo era averaged) geographic locations and depths of isopycnals, with a constant volume being used to integrate the warming effects of each water-mass layer and regionally defined water mass over the Argo period.”, and on lines 497–505: “By decomposing the OHC change into its volumetric and warming components (Eq. 2), the warming component of

OHC change within the water-mass layers and in regionally defined water masses (seen in Figs. 3, 4, 10, and Supplementary Tables 2, 4) is estimated based on a set of fixed longitudes, latitudes, and isopycnal depths (referred to as “fixed volume” or “volume effects removed” in this study) averaged over the Argo era, while the volumetric component is based on a set of density ranges whose depths and volume can evolve over time. This decomposition allows us to distinguish OHC changes of a water mass due to warming effects separate from isopycnal layer thickness (or volume) changes.”.

We have also estimated OHC trends per decade, for the upper 2000 m of the ocean and for the mode and intermediate water layers respectively, based on a set of decadal averaged density fields from 1960 to 2020, as per the reviewer suggestion. This initial test indicates that heat accumulation by the global mode and intermediate water layers can account for 81% and 77% of the total heat accumulation in the upper 2000 m of the ocean over 2000–2010 and 2010–2020, respectively. These values are thus close to the estimate that ~76% of ocean heat accumulation is accounted for by the net warming (fixed volume) of mode and intermediate water layers over the Argo era (see added text on lines 150–157).

5. It is not good practice to name time periods after people. Please find other names than “The Roemmich period” or “The Levitus period”.

Response: Agreed – thanks for pointing this out. We have now referred to the period 1955–2004 as “pre-Argo period” and the period 2005–2020 as “Argo era” in the manuscript. The terminologies of “Levitus period” and “Roemmich period” in the previous version of the manuscript have now all been removed.

6. As presented, the units of J yr^{-1} in Figures 3, 4, 5a, 5c-d, 6, 7, and 8 are all resolution dependent, and need to include area units for the maps (e.g., $\text{J m}^{-2} \text{ yr}^{-1}$) and a length unit for the line plots in Figure 4 ($\text{J m}^{-1} \text{ yr}^{-1}$). Also, the units of Figure 2f are also resolution dependent as presented, and need to be in $\text{J kg}^{-1} \text{ m}^3$.

Response: Thanks for pointing this out. We have now corrected the units in Figs. 2–10 to be resolution independent units, such as [$J m^{-2}$] for Figs. 5a-b, [$J \text{ per degree latitude}$] for Figs. 5c-d, and [$J m^{-2} yr^{-1}$] for Figs. 6a and 7, 9–10. We have also added a clarification to indicate that the unit of Figs. 2f and 3e-f is in [$J \text{ per } 0.1 kg m^{-3} \text{ density bin}$] in the caption of Figs. 2 and 3.

7. Lines 78-89 and elsewhere: There are a number of different groups estimating ocean warming as is documented in von Shuckman et al. (2020, Earth Syst. Sci. Data), where the acceleration noted here is also discussed in some detail. In addition Loeb et al. (2021, Geophys. Res. Lett.) also document the acceleration of ocean warming 2005-2019, and show that it agrees with increases in net-top-of-the atmosphere energy fluxes observed by satellite.

Response: Thanks for the suggestions. We have now added a citation to von Shuckman et al. (2020), the latest assessment report of the IPCC Working Group I, and Loeb et al. (2021) in the manuscript text, and also included clarifications of this point on lines 48–50: “Due to the accumulated excess heat in ocean basins, an acceleration of total ocean warming has become more evident from recent observational-based studies^{5,6,15,16}.”, lines 105–107: “Although some discrepancies of the global OHC estimates are evident among the different datasets, the ensemble mean of OHC estimates for the upper 2000 m shows that ocean warming is not only continuing, but accelerating^{5,8,16} (Fig. 1).”, and lines 114–123: “Further comparison with previous estimates of OHC trends (Table 1) and the latest assessment report of the Intergovernmental Panel on Climate Change (IPCC) Working Group I⁷ also reveals significant increases in the ocean warming rate across the measurement record^{5,8-11,16,17,19}. This increasing ocean warming rate is not just limited to the upper layer of the ocean, but also occurs at intermediate depths (700–2000 m) according to a further breakdown of OHC trends by depth over the upper 2000 m^{5,7}. Significant ocean warming and accelerating OHC changes are consistent with the increase in net radiative energy absorbed by Earth detected in satellite observations^{5,33}, something that is likely to continue throughout the 21st Century^{6,15} in the absence of substantial

greenhouse gas emissions reductions.”.

8. Lines 104-108. Does this perhaps signal actually a change in the density of the mode waters (becoming lighter with time in a warming ocean), rather than a change in volume of intermediate versus mode waters?

Response: Yes, warming of the mode waters can contribute to its volume increase due to changes in the density field. Volume of the intermediate water layer has been decreasing at a rate of $-0.4 \times 10^{14} \text{ m}^3 \text{ year}^{-1}$, and the mode water layer volume has been increasing at a rate of $1.9 \times 10^{14} \text{ m}^3 \text{ year}^{-1}$ over the Argo era 2005–2020 (Figs. 10a-c). By decomposing the temporal OHC change into volumetric and warming (fixed volume) components (Eq. 2), we find that the increased OHC of the mode water layer is due to both net warming (fixed volume) and a volume increase of the layer, while decreased OHC in the intermediate water layer is mainly due to its volume loss (Figs. 3c-f, Table S2). Both the mode and intermediate water layers have shown a near-continuous warming (at fixed depths) that can account for about 76% of the total ocean warming measured across the Argo era (Figs. 3a, c-e, Table S2). We have now included a plot of this point into the main manuscript as Fig. 3, and have rephrased these sentences on lines 148–157: “These opposing signals are mainly due to the mode water layer accumulating heat content via increasing its volume, while the intermediate water layer loses heat content due to volume decreases (Figs. 3c-f). After factoring out these volumetric effects, both the global mode and intermediate water layers have shown a near-continuous and monotonic warming across the Argo era (Figs. 3 and 4), revealing a robust footprint of global warming penetrating into the ocean interior. Taken together, the total heat accumulated in the mode and intermediate water layers is 89% of the net global ocean warming during the Argo era, despite their total volume just occupying 62% of the upper 2000 m. The net warming of these two layers, separate from volume effects, accounts for ~76% of this global ocean warming, while ~12% is due to their combined volume change (Fig. 3a, Supplementary Table 2).”.

We have also included a statement to clarify the uncertainty of using a fixed density range across the Argo era at lines 319–325 and in *Methods* (lines 491–505).

9. Line 125. Here it would be appropriate to note the work of Roemmich, who noted previously on that recent warming has been concentrated in the southern Ocean.

Response: Yes. Done – thanks.

10. Lines 170-186. This explication (and others elsewhere which include volume changes) really overemphasizes the importance of warming in the mode and intermediate waters. Volume changes can result in huge “heat” changes, but since heat changes really should be evaluated in a constant volume, they don’t mean much and need to be explained very carefully.

Response: Thanks for pointing this out - agreed. To separately address the role of warming (fixed volume) and volumetric components in total OHC changes, we have now rephrased the statements in this section (now on lines 226–238): “The total heat content of a water mass can increase both because it warms, or because the total volume of that water mass increases, or both (Methods). By decomposing the OHC increase over the Argo era 2005–2020, we find that the actual warming (within a fixed volume) of all regionally defined mode and intermediate waters can account for 48% of the global ocean warming (Fig. 9a, 10g; Supplementary Tables 3, 4). The volume of these water masses also changes markedly but non-uniformly across this period, with their net volume change overall explaining a further –4% of the global OHC increase. The combined OHC increase of the sum of all these regionally defined mode and intermediate waters, due to both warming (fixed isopycnal depth and volume) and volume changes, amounts to a total contribution of 44% to the global OHC increase (using the values shown in Fig. 9a and Supplementary Table 4). The total volume of those regionally defined water masses within the subtropical and subpolar oceans occupies just 24% of the upper 2000 m averaged over 2005–2020.”. Text on this point has also been rephrased at lines 148–157, which can be seen in our response to comment #8 above.

We have also now added clarifications such as “fixed volume” and “volume effects removed” when we refer to the contribution from warming of water masses on lines 156, 228, 234, 252, 270, and 314, and removed the sentence on lines 170–172 of the original manuscript, to avoid overemphasizing the role of warming in mode and intermediate waters. The warming component of OHC change (seen in Figs. 3–4, 10, and Tables S2, S4) is estimated based on a set of fixed longitudes, latitudes, and isopycnal depths averaged over the Argo era (Eq. 2), and therefore, a set of constant volumes have been used to integrate the warming effects of each water mass. We have added a statement on this point at lines 433–438 and 497–505.

REVIEWER COMMENTS

Reviewer #1 (Remarks to the Author):

It is nice that the authors have addressed most of my previous comments. I have some follow-on suggestions, which are not substantial, mainly help the authors to correct some minor errors and improve the presentation of the manuscript, so I'd classified my remaining comments as "minor revisions". I hope my suggestions (sometimes very detailed and specific) could help the authors to improve the paper and make it more powerful and impactful.

- 1) Line 87-138: this section can be shortened. Many of the texts within line 124-135 can be moved into the Methods section when justifying your data/method.
- 2) Line 90: "since the late 1970s", should be "since the late 1960s"
- 3) Line 99: "reanalysis", should change to "objective analysis"
- 4) Line 105: "discrepancies" better for "differences"
- 5) Line 93-95: ref.5 did not do this, you should cite ref.6 here.
- 6) Line 107: here ref.8 does not provide any analysis for rate change, you should replace ref.8 by ref.15 here, for the very first claim of warming acceleration.
- 7) Fig.1, a very important figure of this study, can be improved. 1), the terminology "ocean heat accumulation" should be changed or clearly explained (please check Cheng et al. 2022 NREE paper for terminology related to ocean warming, this term of "ocean heat accumulation" is rarely used and not clear). 2) what are the blue rectangles? Why the central points of the blue bars are located at the top of the rectangles? These are not clear. Please make it a self-explanatory figure, which is very helpful for readers to quickly get your points. 3) in panel-a, the start and end points of the three periods are not apparent (one needs to read the figure caption to know this), probably you can add light (maybe different level of grey) background shadings to better illustrate the three periods.
- 8) Fig. 1b, Cheng et al. 2022 J. Climate indicates that linear trend is only valid for ~15 years segments, thus please consider change your segments accordingly to make the plot more meaningful.
- 9) I strongly object using EN4, as explicitly stated regarding the EN4 data paper Good et al. 2013: "It is important to note that the analyses will relax to climatology in the absence of any observations. Care must therefore be taken if using the analyses for applications such as identifying trends in temperature or salinity, because a trend may be unrealistic if analyzing periods when there were no observations.". It is not recommended to use EN4 for trend analysis as the current study. Furthermore, if you look at Supplementary Fig.1, the big shift around 2000 is not physical, and the changes before 2000 are too little to be true.
- 10) Fig. 4: can the red bars be more transparent so one can see the dark blue bars behind?
- 11) Line 172-183 and Fig.5, I actually don't find these discussions needed in this paper. The difference of the trends for the two periods are unclear, as you correctly stated in line 187-188 "Note that trends within this relatively short period can indicate both the forcing signal due to anthropogenic warming as well as signals superimposed due to internal variability", the "regional intensification" of trend for 2010-2020 compared with 2000-2010 period is not very meaningful and out of the scope of this study (stick to the OHC changes in different water masses would be interesting enough, and stick to 2005-2020 would be sufficient).
- 12) Fig. 8, probably you can merge this figure with Fig.6, to make each plot more informative.
- 13) Fig. 8, are you able to better separate SP-ESTMW and SP-WSTMW, NP-CMW and NP-STMW in the plot? Where are the edges of these water masses?

- 14) Fig. 6, please consider moving panel-d into the SI, because it is not directly linked to the main flow.
- 15) Fig. 9 caption, you don't need to repeat the full name of all water masses.
- 16) Line 243: not all water masses are warming and thickening, so please consider revise the title.
- 17) Line 308: this line probably starts a new section of "Summary and Discussion"?
- 18) Line 337-340: this has repeated what you have said before. This paragraph needs to be shortened.
- 19) Line 347-351: this should be shown combined of the quantitatively statements within 308-318.
- 20) Lin 351-353: in previous paragraph, you already stated something similar about "better measure of heat". Please improve the organization of the final two paragraphs.
- 21) Line 356-357: "a total of 8": different members of EN4 should be regarded as 1 dataset.
- 22) Line 363: "relatively minor", please consider to change it to a more objective presentation, maybe "better consistency over XX than YY".
- 23) Line 366-369: this choice means the global OHC estimate will be biased because of the smaller area.
- 24) Line 382-384: IAP salinity is "absolute salinity", so you should not need to do conversion for this dataset.
- 25) Line 419-421: just to check: is this following TEOS-10 standard?
- 26) Line 507-512: how did you consider the reduction in degree of freedom when calculating least-square linear trend?
- 27) Line 540-542, IAP data link is: <http://www.ocean.iap.ac.cn/>
- 28) Table 1. Because you aim to show the "accelerating global ocean warming", probably you can rearrange the order of rows, starting from longest-period trend and end up with most-recent-period trend, in this way, one can nicely observe the increase in trends from up to bottom.
- 29) Figure 10: Please consider to make it similar to the style of Fig. 9. At least, you can make bar plots at top and then spatial maps below, so one can better compare Fig.9 and Fig.10. To further improve the illustration, Fig.9 and 10 can be combined into a powerful/informative plot (deserve to do so for this high-level journal).
- 30) Fig. 3, the x-axis is not consistent between the first four panels and the last two, please make it consistent, so one can better compare upper four panels with the lowest two.
- 31) Fig. 4: Actually I think you can combine Fig.4 into Fig.3, by make Fig.4 small and put it to the right side the Fig.3f. Actually in Fig.4, doing the statistics for every 0.1 kg m^{-3} is not necessary, you can do it for 0.5 (or 0.25 at least), to simplify the plot without losing the key information you want to convey.

Reviewer #2 (Remarks to the Author):

General Comments.

As I said in the first review "I have enjoyed reading this paper. It has a wide appeal to oceanographers and to a wider communications audience. This appeal is because the manuscript documents in a more thorough way than previously the changes in a key water-mass type that is found widely across the global oceans. The global perspective of this paper means that it details a coherent set of changes in both northern and southern hemispheres." and that "My belief is that this is a very relevant and timely paper for nature communications.

“

The reviewers have responded to my comments very thoroughly. They have adopted an ensemble approach through out all of the figures, provided estimates of error bars in figures and tables.

From my perspective they have thoroughly addressed my original comments.

The only quibble I now have is the use of the word acceleration in heat content. The acceleration that is presented is often based on differing periods of time, and trends calculated over shorter and long periods will always be different. Thus this assertion of an acceleration can really only be made when the trend periods are presented for the same length of time. The text should be altered to be clearer on this point.

Reviewer #3 (Remarks to the Author):

This manuscript analyzes ocean heat content using eight (slightly) different g time-series of global ocean heat content and the mean of three (slightly) different gridded maps for ocean temperature and salinity fields. It focuses on contributions of mode and intermediate water warming to ocean warming as a whole. The revision is improved from the original, but the one of main issues with the manuscript brought up in the previous review remain largely unaddressed, and another is not adequately addressed in the revision.

1. The potential impact of increases in ocean sampling on the acceleration of ocean warming between the 1990s and 2010-2020 remains under-acknowledged. For instance, the claim of doubling rate over those time periods is made in the abstract without any mention of possible mapping artifacts. After a full paragraph (L105-121) on the acceleration, only one oblique sentence (l124-126) briefly mentions this issue. Linking to acceleration of sea level rise might provide a bit more context.

2. Second, the manuscript still mis-interprets the volume changes in isopycnal classes For instance, the long and elaborate description of possible reasons for changes in L261-293 glosses over the simplest explanation, that warmer surface conditions propagate down, and density horizons shift for the various water masses. What was once a subtropical mode water density class has shifted towards what is now a subpolar mode water density class and what was once a subpolar mode water density class shifts to an intermediate water density class. This interpretation is very clear in the figures, and pretty clear in the previous literature on the subject. For instance, look as the quadrupoles at the boundaries between those water masses in Figure 2f and you can see the all the water masses shifting to lighter density classes. The same effect can be seen even more clearly in Figure 3f. Figures 9d and 9e also show these changes really clearly, with the SMPW supposedly shifting southward and eastward in the Indian and Southwest Pacific Ocean, and intermediate water supposedly disappearing in the in the Southeast Pacific Ocean. It seems much more likely (and supported by other analyses in the literature) that the density of intermediate water has lessened over time, as has the density of SPMW, and STMW, as the surface conditions have warmed (and in some cases freshened). This is reflected in the change of volumes in fixed density layers, as the water masses themselves shift to lighter layers.

3. This a more minor point, but on L218-220 and elsewhere the manuscript seem to grapple with how heat builds up in the mode and intermediate waters. These waters warm mostly because surface waters warm, even in winter, and thus the mode and intermediate waters warm when wintertime mixing deepens the reach of that warmer layer.

RESPONSE TO REVIEWER COMMENTS:

Below reviewer comments have been copy-pasted in black font, and our responses are indicated in blue font. Line numbers in our responses refer to the revised manuscript (except where indicated).

Reviewer #1 (Remarks to the Author):

It is nice that the authors have addressed most of my previous comments. I have some follow-on suggestions, which are not substantial, mainly help the authors to correct some minor errors and improve the presentation of the manuscript, so I'd classified my remaining comments as "minor revisions". I hope my suggestions (sometimes very detailed and specific) could help the authors to improve the paper and make it more powerful and impactful.

We thank Reviewer #1 for this favourable assessment of our paper, and for their insightful and detailed comments in this round of review. We believe these comments have significantly helped us to improve the paper. Thank you.

Reviewer comments:

1) Line 87-138: this section can be shortened. Many of the texts within line 124-135 can be moved into the Methods section when justifying your data/method.

Response: Agreed - thanks for the suggestions. We have now rephrased the text on increasing ocean warming rate in this section (lines 109–157 of the revised manuscript),

and have moved the text on lines 125–131 into the *Methods* section and rephrased these sentences as (lines 373–379):

“Several factors can affect the estimate of ocean warming rate and should be factored in, particularly over the pre-Argo period (Fig. 1, Supplementary Fig. 1). Such factors include uncertainties due to data scarcity and data quality prior to the late 1960s¹⁰, mapping artifacts from horizontal and vertical interpolation of the sparse historical data⁶³, the increased spatial sampling coverage over time (particularly in the tropical and Southern Hemisphere oceans), and the shift of the observational network from a ship-based system to the Argo network during 2001–2003³³.”,

and on lines 530–533 of the revised manuscript:

“For linear trends within a period less than 15 years, decadal variability can be part of the signal as well as long-term trends due to anthropogenic warming¹⁶. Another factor is start- and end-point sensitivity in estimating linear trends and capturing nonlinearity in rates of OHC change¹⁶. Overall, the monotonic increase in ocean heat uptake remains clear since the 1980s.”.

2) Line 90: “since the late 1970s”, should be “since the late 1960s”

Response: Thanks for the suggestion. Here in the manuscript, we want to highlight the period when XBTs were *widely* used for observing the upper ocean. We have now changed the text to “..., which have been widely used since the 1970s” because the deployment of XBTs began in the mid to late 1960s, and XBTs thereafter became the largest source of data for the upper ocean thermal record during the 1970s–1990s (Goni et al. 2019), but were only widely used from the 1970s onwards.

3) Line 99: “reanalysis”, should change to “objective analysis”

Response: Done – thanks.

4) Line 105: “discrepancies” better for “differences”

Response: Agreed and changed. Thanks for the suggestion.

5) Line 93-95: ref.5 did not do this, you should cite ref.6 here.

Response: Done – thanks.

6) Line 107: here ref.8 does not provide any analysis for rate change, you should replace ref.8 by ref.15 here, for the very first claim of warming acceleration.

Response: Agreed - done. Thanks

7) Fig.1, a very important figure of this study, can be improved. 1), the terminology “ocean heat accumulation” should be changed or clearly explained (please check Cheng et al. 2022 NREE paper for terminology related to ocean warming, this term of “ocean heat accumulation” is rarely used and not clear). 2) what are the blue rectangles? Why the central points of the blue bars are located at the top of the rectangles? These are not clear. Please make it a self- explanatory figure, which is very helpful for readers to quickly get your points. 3) in panel-a, the start and end points of the three periods are not apparent (one needs to read the figure caption to know this), probably you can add light (maybe different level of grey) background shadings to better illustrate the three periods.

Response: Agreed - good suggestions, thanks. We have now changed the terminology “ocean heat accumulation” to “ocean heat uptake” in Fig. 1b and updated all related text

in the manuscript accordingly. The blue rectangle bars in Fig. 1b represent the global ocean heat uptake for every 11-year period (e.g. 1990–2000, 2000–2010, and 2010–2020), which is calculated as the linear trend of annual mean OHC multiplied by the length of the period; superimposed blue error bars in Fig. 1b indicate the uncertainty range of global ocean heat uptake (blue rectangle bars) from various products. We have now reworded text on this point in the caption of Fig. 1 to be clearer on this:

“**b**, Blue rectangle bars indicate the ensemble averaged global ocean heat uptake (10^{22} J) for every 11-year period across the measurement era (Methods). Superimposed error bars indicate the ± 2 ensemble standard deviation uncertainty range ($\pm 2\sigma$) of global ocean heat uptake across various datasets.”.

We have also added delineating arrows into Fig. 1a to better indicate the start and end years of the pre-Argo period, WOCE era, and Argo era, respectively, as per the reviewer suggestion.

8) Fig. 1b, Cheng et al. 2022 J. Climate indicates that linear trend is only valid for ~15 years segments, thus please consider change your segments accordingly to make the plot more meaningful.

Response: Thanks for the suggestion. We agree that the OHC trends over each 11-year period might include both decadal variability and the long-term change. Overall, the increase in OHC remains clear across the measurement era, particularly since the 1980s. We did not change the 11-year ocean heat uptake in Fig. 1b to over a 15-year segment. Instead, we have added text on this point at lines 530–533:

“For linear trends within a period less than 15 years, decadal variability can be part of the signal as well as long-term trends due to anthropogenic warming¹⁶. Another factor is start- and end-point sensitivity in estimating linear trends and capturing nonlinearity in rates of OHC change¹⁶. Overall, the monotonic increase in ocean heat uptake remains clear since the 1980s.”

9) I strongly object using EN4, as explicitly stated regarding the EN4 data paper Good et al. 2013: “It is important to note that the analyses will relax to climatology in the absence of any observations. Care must therefore be taken if using the analyses for applications such as identifying trends in temperature or salinity, because a trend may be unrealistic if analyzing periods when there were no observations.”. It is not recommended to use EN4 for trend analysis as the current study. Furthermore, if you look at Supplementary Fig.1, the big shift around 2000 is not physical, and the changes before 2000 are too little to be true.

Response: Thanks for pointing this out. We agree that there are large inconsistencies within the four ensembles of the EN4 data over the 1990s and a large shift of global OHC around 2000 (Supplementary Fig.1), which gives us limited confidence in any analysis from the EN4 data over the pre-Argo period. Therefore, we have now excluded the EN4 data over the pre-Argo period 1955–2004 and have only included the EN4 data over the Argo period 2005–2020 in this study. We have now updated Fig. 1 and Fig. 5 (Fig. 2 of the revised manuscript), and all related text in the manuscript. We have also added a statement of this point in the *Methods* section (lines 398–401):

“Due to the large inconsistency within the four EN4.2.2 ensembles over the 1990s, and due to a large shift of global OHC around 2000–2004 in the EN4.2.2 ensemble mean, we have only included the EN4 data over the Argo period 2005–2020 in this study.”.

10) Fig. 4: can the red bars be more transparent so one can see the dark blue bars behind?

Response: Agreed and changed. Thanks for the suggestion.

11) Line 172-183 and Fig.5, I actually don't find these discussions needed in this paper. The difference of the trends for the two periods are unclear, as you correctly stated in line 187-188 "Note that trends within this relatively short period can indicate both the forcing signal due to anthropogenic warming as well as signals superimposed due to internal variability", the "regional intensification" of trend for 2010-2020 compared with 2000-2010 period is not very meaningful and out of the scope of this study (stick to the OHC changes in different water masses would be interesting enough, and stick to 2005-2020 would be sufficient).

Response: Thanks for pointing this out. We have now moved Fig. 5 and Fig. 8 to the section "*Accelerating global ocean warming*" to Fig. 2 and Fig. 3 in the revised manuscript, respectively. We have also moved the text on lines 172–183 to the section "*Accelerating global ocean warming*", and reworded this text to better indicate the spatial variability of ocean warming in the past few decades (lines 134–150 of the revised manuscript):

"The increased ocean warming is non-uniformly distributed across ocean basins. Overall, in each ocean basin, an increase in OHC is observed (values indicated in Figs. 2a, b), with stronger warming in the mid-latitude Atlantic Ocean and the Southern Ocean compared with other basins^{6,11}. Total warming in the Southern Ocean is estimated to account for ~31% of the global upper 2000-m OHC increase from 1980–2000 to 2000–2010 (Fig. 2a), and almost half of the global OHC increase from 2000–2010 to 2010–2020 (values indicated in parentheses of Fig. 2b). Hence the Southern Ocean has seen the largest increase in heat storage over the past two decades, holding almost the same excess anthropogenic heat as the Atlantic, Pacific, and Indian Oceans north of 30°S combined (Fig. 2d). The most striking warming in the Southern Ocean is concentrated on the northern flank of the Antarctic Circumpolar Current, the location of deep mixed layers and subduction hotspots for Subantarctic Mode Water and Antarctic Intermediate Water, as well as the location of subtropical mode waters formation further equatorward (Fig. 3). The well-ventilated regions near western boundary current extensions in the North Atlantic and North Pacific also reveal large warming over the

past two decades. These hotspots of ocean warming are likely linked to enhanced uptake, subduction, and lateral spreading of heat associated with mode and intermediate waters that warrants further investigation.”.

12) Fig. 8, probably you can merge this figure with Fig.6, to make each plot more informative.

Response: Thanks for the suggestion. We have now moved Fig. 8 to the section “*Accelerating global ocean warming*” as Fig. 3 in the revised manuscript, to better address the spatial variability of ocean warming. In this case, we did not merge Fig. 8 with Fig. 6, in part due to the new figure location, and also to avoid overcrowding the diagram.

13) Fig. 8, are you able to better separate SP-ESTMW and SP-WSTMW, NP-CMW and NP-STMW in the plot? Where are the edges of these water masses?

Response: Thank you for the suggestion. We have now updated Fig. 8 (Fig. 3 of the revised manuscript) by adding the thickness contours of all Subtropical Mode Waters, NP-CMW, NP-IW, NA-SPMW, SAMW, and AAIW, and into the plot. The thickness of NA-MMW is not included in Fig. 3 for simplicity and clarity of this plot. Superimposed dashed contours in the new Fig. 3 indicate the 150-m, 200-m, 250-m, 550-m, 650-m, and 1000-m thickness of water masses based on the density constraints in Supplementary Table 3, and are represented by different colours. Now the geographic locations of all these water masses, including SP-ESTMW, SP-WSTMW, NP-CMW, and NP-STMW, can be better indicated within the plot. Text on this point has also been added to the caption of Fig. 3:

“Superimposed contours represent the 150-m, 200-m, and 250-m thickness of all STMWs and NP-CMW (blue), 550-m thickness of NP-IW (red), 650-m thickness of

SAMW and NA-SPMW (green), and 1000-m thickness of AAIW (yellow), to indicate their geographic locations. Note that the water-mass thickness is estimated based on the density constraints given in Supplementary Table 3, and for simplicity, only the core thickness of each water mass is shown. The geographic constraints for estimating OHC change of water masses are referred to in Supplementary Table 3.”.

14) Fig. 6, please consider moving panel-d into the SI, because it is not directly linked to the main flow.

Response: Thanks for the suggestion. We considered moving this panel into the supplementary material, but that file will be separated from the main published paper. We feel that keeping it in the main manuscript would give readers a straightforward view of the effects of ocean thermal expansion and haline contraction in the ocean steric sea-level rise.

15) Fig. 9 caption, you don't need to repeat the full name of all water masses.

Response: Agreed and changed. Thanks for the suggestion. We have now only kept the full name of water masses in the caption of Fig. 3 and only the acronym of water-mass names in the caption of Fig. 9.

16) Line 243: not all water masses are warming and thickening, so please consider revise the title.

Response: Thanks for the suggestion. We considered changing the title of this section, but we feel that the original title summarizes the section in a nice succinct way as most of the selected subtropical mode waters and SAMW have been warming and thickening, while the warming of AAIW and NPIW has also been substantial across the Argo period.

We therefore decided to keep the previous title for this section. Since we have reworded the second paragraph of this section (lines 273-286) as per the comment from Reviewer #3, the revised text for this section now fits better to the previous title.

17) Line 308: this line probably starts a new section of “Summary and Discussion”?

Response: Agreed and changed – thanks.

18) Line 337-340: this has repeated what you have said before. This paragraph needs to be shortened.

Response: Thanks for raising this. We have now rephrased and shortened the “*Summary and discussion*” section, and combined the text on lines 337–340 with the first paragraph of this section (lines 319–336 of the revised manuscript) to:

“The world’s oceans hold by far the largest excess anthropogenic heat in Earth’s climate system¹. Knowing exactly where and how much heat is stored in the ocean is thus fundamental to our ability to understand and predict future climate change and sea-level rise. In addition, this knowledge can help inform how to optimise observing networks for monitoring the state of the ocean and climate system. Our study has documented a strong acceleration in global ocean warming since the 1990s, amounting to >25% increase in OHC during 2010–2020 relative to 2000–2010, and nearly a twofold increase during 2010–2020 relative to 1990–2000. Approximately 89% of the increase in global ocean heat uptake, which includes warming and volume changes, is confined to global mode and intermediate water density layers during the Argo era 2005–2020. Mode and intermediate waters play a vital role in absorbing and redistributing tracers such as heat, carbon, and oxygen in the global ocean. The warming of 13 of these regionally defined mode and intermediate waters at fixed geographic locations and isopycnal depths (i.e. with volume effects removed) accounts for ~48% of global ocean warming despite their volume occupying just 24% of the

ocean. This includes Subtropical Mode Waters in both hemispheres, which have overall shown a striking increase in OHC dominated by increases in volume, and Subantarctic Mode Water and Antarctic Intermediate Water in the Southern Ocean, which have warmed substantially and are responsible for more than one-third of global ocean warming across the 2005–2020 period.”.

19) Line 347-351: this should be shown combined of the quantitatively statements within 308-318.

Response: Thanks for the suggestions. We have now combined the text on lines 347–351 with the previous paragraph and rephrased the text into a new paragraph (lines 319–336 of the revised manuscript), which can be seen in our response to comment #18 above.

We have also rephrased the last paragraph of the “*Summary and discussion*” section to (lines 355–363 of the revised manuscript):

“Our work reveals accelerated warming in the upper 2000 m of the ocean over the past several decades and highlights increasing heat uptake by mode and intermediate waters, with these two water masses responsible for the majority of ocean warming over the Argo era (2005-2020), despite a limited area of interaction with the atmosphere. Exactly how this heat uptake plays out over the coming decades and beyond remains highly uncertain. For example, climate change induced warming and freshening at the surface is projected to stratify the upper ocean, which will reduce the overturning of these water masses, in turn reducing their capacity to uptake heat. This would have profound implications for the rate of future anthropogenic climate change.”.

20) Line 351-353: in previous paragraph, you already stated something similar about “better measure of heat”. Please improve the organization of the final two paragraphs.

Response: Thanks for pointing this out. We have now removed this sentence on lines 351–353, and have combined this paragraph with the previous paragraph (lines 319–336 of the revised manuscript), which can be seen in our response to comment #18 above. We have also rephrased the last paragraph of the “*Summary and discussion*” section (lines 355–363 of the revised manuscript), which can be seen in our response to comment #19 above.

21) Line 356-357: “a total of 8”: different members of EN4 should be regarded as 1 dataset.

Response: Agreed and changed. Thanks for the suggestion.

22) Line 363: “relatively minor”, please consider to change it to a more objective presentation, maybe “better consistency over XX than YY”.

Response: Thanks for pointing this out. We have now revised this statement to (lines 371–373): “The estimate of global and basin-wide OHC trends among these selected observational-based products were previously shown to exhibit good agreement in the upper 2000-m ocean over the Argo era⁴⁴ (Fig. 1).”.

23) Line 366-369: this choice means the global OHC estimate will be biased because of the smaller area.

Response: Yes, agreed. Therefore we have added a statement on this point in lines 385–387: “As such, the resultant global OHC trends from SIO RG-Argo in Fig. 1 will be underestimated if these excluded polar ocean regions are also warming⁵.”.

In Figs. 2–10 of the manuscript, where we show OHC changes within ocean basins and in mode and intermediate waters, we focus on the region between 65°S–65°N due to the low sampling coverage at higher latitudes and for better consistency of the data’s spatial coverage. Text on this point has now been added into the caption of Fig. 2, and can also be seen in the caption of Figs. 4, 5 and Supplementary Table 1, and on lines 154–157 in the revised manuscript: “Thus, for the remainder of this study we focus on the Argo era 2005–2020, a period common to all data products analyzed. We also limit the water-mass decomposition to be between 65°S and 65°N, because of sparse observations in the polar regions even after the Argo array was established.”.

24) Line 382-384: IAP salinity is “absolute salinity”, so you should not need to do conversion for this dataset.

Response: Thanks for pointing this out. We have now corrected this statement as “The Gibbs-Sea Water (GSW) Oceanographic Toolbox from the TEOS-10 software^{64,65} is applied to convert the SIO RG-Argo data and EN4.2.2 data to Conservative Temperature⁶⁶ and Absolute Salinity⁶⁷, as well as the IAP *in-situ* temperature to Conservative Temperature, ...”.

25) Line 419-421: just to check: is this following TEOS-10 standard?

Response: Yes, all variables in Eq. (1) were calculated following TEOS-10 standards, where c_p is a constant of $3992 \text{ J kg}^{-1} \text{ K}^{-1}$, ρ is potential density in kg m^{-3} , and θ is the potential temperature with respect to $0 \text{ }^\circ\text{C}$ and is measured in degrees Celsius.

26) Line 507-512: how did you consider the reduction in degree of freedom when calculating least-square linear trend?

Response: Thanks for pointing this out. Actually, we separately calculate the linear trend over each selected period (e.g., 1955–2004, 2005–2020, 2010–2020), with the degree of freedom chosen as 1 for the least-squares regression. We have now removed “using a linear least-squares regression” to avoid any confusion and revised that sentence as (lines 523–527 of the revised manuscript):

“Linear trends of OHC are calculated within each selected time period for spatially integrated annual mean OHC time series, to indicate the increase or decrease in OHC across each selected period. For the trend maps, the fields are first vertically integrated over depth or isopycnal layers, and then the linear trends over the selected period are computed for each grid point.”.

27) Line 540-542, IAP data link is: <http://www.ocean.iap.ac.cn/>

Response: Thanks and changed.

28) Table 1. Because you aim to show the “accelerating global ocean warming”, probably you can re-arrange the order of rows, starting from longest-period trend and end up with most-recent-period trend, in this way, one can nicely observe the increase in trends from up to bottom.

Response: Yes. Done – good suggestion, thanks.

29) Figure 10: Please consider to make it similar to the style of Fig. 9. At least, you can make bar plots at top and then spatial maps below, so one can better compare Fig.9

and Fig.10. To further improve the illustration, Fig.9 and 10 can be combined into a powerful/informative plot (deserve to do so for this high-level journal).

Response: Agreed - thanks for the suggestions. We have now updated Fig. 9 and made Figs. 9 and 10 in the same style, i.e., both with spatial maps at the top and bar plots at the bottom. We also carefully considered combining Fig. 9 with Fig. 10, however, it came out too crowded for the combined new figure. So we have decided to keep them as two separate plots in the revised manuscript.

30) Fig. 3, the x-axis is not consistent between the first four panels and the last two, please make it consistent, so one can better compare upper four panels with the lowest two.

Response: Agreed and changed. Thanks for the suggestion.

31) Fig. 4: Actually I think you can combine Fig.4 into Fig.3, by make Fig.4 small and put it to the right side the Fig.3f. Actually in Fig.4, doing the statistics for every 0.1 kg m⁻³ is not necessary, you can do it for 0.5 (or 0.25 at least), to simplify the plot without losing the key information you want to convey.

Response: Thanks for the suggestions. We looked at combining these two figures, however, it came out too crowded for the combined new figure. In this way, we have decided to keep Figs. 3 and 4 as two separate plots (Figs. 5 and 6 in the revised manuscript), and Fig. 4 (now Fig. 6) shown for every 0.1 kg m⁻³ density bin for consistency with Figs. 2 and 3 (now Figs. 4 and 5).

Reviewer #2 (Remarks to the Author):

General Comments.

As I said in the first review “I have enjoyed reading this paper. It has a wide appeal to oceanographers and to a wider communications audience. This appeal is because the manuscript documents in a more thorough way than previously the changes in a key water-mass type that is found widely across the global oceans. The global perspective of this paper means that it details a coherent set of changes in both northern and southern hemispheres. “ and that “My belief is that this is a very relevant and timely paper for nature communications.”

The reviewers have responded to my comments very thoroughly. They have adopted an ensemble approach through out all of the figures, provided estimates of error bars in figures and tables.

From my perspective they have thoroughly addressed my original comments.

We thank Reviewer #2 for this favourable assessment of our paper, and for their insightful comments in these two rounds of review. We believe these comments have significantly helped us to improve the paper. Thank you.

The only quibble I now have is the use of the word acceleration in heat content. The acceleration that is presented is often based on differing periods of time, and trends calculated over shorter and long periods will always be different. Thus this assertion of an acceleration can really only be made when the trend periods are presented for the same length of time. The text should be altered to be clearer on this point.

Response: Agreed - thanks for pointing this out. To address this, when indicating the increase in ocean warming over different time periods in Table 1 and over the pre-Argo and Argo periods in Fig. 1a, we have converted the units to J/year to take into account the different lengths of the different periods. The quantification and description of the increased ocean warming rate over the past three decades in Fig. 1b are based on the same length of periods over 1990–2000, 2000–2010, and 2010–2020. Therefore, the term “acceleration” is appropriate when describing the increase in ocean warming. We have now adjusted the statements on lines 109–133 and rearranged Table 1 to better address this point.

We further agree that the choice of different lengths of time periods and start- and end-points can also affect the estimate of linear trends due to the nonlinearity in ocean warming and internal climate variability. For linear trends within a shorter record, interannual and decadal variability of the climate system can have a larger influence. We have now added text on this point at lines 530–533:

“For linear trends within a period less than 15 years, decadal variability can be part of the signal as well as long-term trends due to anthropogenic warming¹⁶. Another factor is start- and end-point sensitivity in estimating linear trends and capturing nonlinearity in rates of OHC change¹⁶. Overall, the monotonic increase in ocean heat uptake remains clear since the 1980s.”.

Reviewer #3 (Remarks to the Author):

This manuscript analyzes ocean heat content using eight (slightly) different g time-series of global ocean heat content and the mean of three (slightly) different gridded maps for ocean temperature and salinity fields. It focuses on contributions of mode and intermediate water warming to ocean warming as a whole. The revision is improved from the original, but the one of main issues with the manuscript brought up in the previous review remain largely unaddressed, and another is not adequately addressed in the revision.

We thank and appreciate Reviewer #3 for the extensive and detailed comments in these two rounds of review, which have helped us to significantly improve the paper.

1. The potential impact of increases in ocean sampling on the acceleration of ocean warming between the 1990s and 2010-2020 remains under-acknowledged. For instance, the claim of doubling rate over those time periods is made in the abstract without any mention of possible mapping artifacts. After a full paragraph (L105-121) on the acceleration, only one oblique sentence (l124-126) briefly mentions this issue. Linking to acceleration of sea level rise might provide a bit more context.

Response: Thanks for pointing this out. We agree that the increase in ocean sampling and the shift of the observing network from a ship-based system to the Argo network in the early 2000s can potentially influence the estimate of ocean warming rate in the past three decades. We have now included a statement on this point to better acknowledge these possible uncertainties, and also added text on the point of acceleration in global mean sea-level rise at lines 123–133:

“Note that there has been both increased ocean sampling and a shift of the observational network from a ship-based system to the Argo network since the initiation of the global Argo array (2001–2003)³³. This may impact the estimated increase in global ocean warming over the past three decades (Fig. 1). However, the rate of global mean sea-level rise has also been increasing since 1993 based on an independent estimate from satellite altimeter³⁴, providing confidence in our results given that sea surface height increases have been dominated by temperature effects since altimeter measurements began. Significant ocean warming and accelerating OHC changes are also consistent with the increase in net radiative energy absorbed by Earth detected in satellite observations^{5,35}, something that is likely to continue throughout the 21st Century^{6,15} in the absence of substantial greenhouse gas emissions reductions.”, and lines 373–379 of the *Methods* section:

“Several factors can affect the estimate of ocean warming rate and should be factored in, particularly over the pre-Argo period (Fig. 1, Supplementary Fig. 1). Such factors include uncertainties due to data scarcity and data quality prior to the late 1960s¹⁰, mapping artifacts from horizontal and vertical interpolation of the sparse historical data⁶³, the increased spatial sampling coverage over time (particularly in the tropical and Southern Hemisphere oceans), and the shift of the observational network from a ship-based system to the Argo network during 2001–2003³³.”.

2. Second, the manuscript still mis-interprets the volume changes in isopycnal classes. For instance, the long and elaborate description of possible reasons for changes in L261-293 glosses over the simplest explanation, that warmer surface conditions propagate down, and density horizons shift for the various water masses. What was once a subtropical mode water density class has shifted towards what is now a subpolar mode water density class and what was once a subpolar mode water density class shifts to an intermediate water density class. This interpretation is very clear in the figures, and pretty clear in the previous literature on the subject. For instance, look at the quadrupoles at the boundaries between those water masses in Figure 2f and you can see all the water masses shifting to lighter density classes. The same effect can

be seen even more clearly in Figure 3f. Figures 9d and 9e also show these changes really clearly, with the SMPW supposedly shifting southward and eastward in the Indian and Southwest Pacific Ocean, and intermediate water supposedly disappearing in the in the Southeast Pacific Ocean. It seems much more likely (and supported by other analyses in the literature) that the density of intermediate water has lessened over time, as has the density of SPMW, and STMW, as the surface conditions have warmed (and in some cases freshened). This is reflected in the change of volumes in fixed density layers, as the water masses themselves shift to lighter layers.

Response: Thanks for pointing this out - agreed. We have now rephrased the text on lines 261–278 (now lines 273–286 of the revised manuscript), to address this point:

“Among the mode and intermediate waters defined in this study, we find that OHC trends of mode waters are dominated by their volumetric changes. Instead warming (fixed volume) exhibits a larger signal in the heat content trend for intermediate waters, including both Antarctic and North Pacific Intermediate Water (Figs. 3, 6 and 10; Supplementary Table 4). This is likely because while the warming of water masses propagates downward, the density-defined boundary of water masses shifts both vertically and laterally as a response to ocean warming (Supplementary Fig. 2). In particular, the subtropical oceans and the Southern Ocean reveal a significant deepening and poleward shift of mode water isopycnals. This indicates that the Subtropical/Subantarctic Mode Water has been expanding and shifting towards locations that once were classified as Subpolar Mode Water and/or intermediate water. Note that although the mode water has shifted, this does not necessarily indicate a net warming of the water mass. Instead, the intermediate water layer could potentially be lightened and warmed due to mixing with the warmed and deepened mode waters over the subtropical oceans and the Southern Ocean¹⁸ (Fig. 10). In reality, a combination of these two effects has likely played out.”

We have also added a plot of linear trends in zonally averaged temperature and density over the Argo era as Supplementary Fig. 2, to better indicate the vertical and lateral shift of density classes due to warming over the subtropical and Southern Oceans.

3. This a more minor point, but on L218-220 and elsewhere the manuscript seem to grapple with how heat builds up in the mode and intermediate waters. These waters warm mostly because surface waters warm, even in winter, and thus the mode and intermediate waters warm when wintertime mixing deepens the reach of that warmer layer.

Response: Thanks for the suggestion and apologies this was not clearer in the revised manuscript. Text on this point has now been added (lines 226–230) to the revised manuscript:

“Therefore, a component of this accelerated OHC increase must be due to increased heat uptake by mode and intermediate waters via wintertime ventilation, in combination with deepening of the warmed water by mixing. In this way warming from the surface can propagate into the ocean interior and spread laterally and vertically via circulation and mixing with surrounding water masses.”.

REVIEWER COMMENTS

Reviewer #1 (Remarks to the Author):

The paper is ready to publish, with only one very minor language editing suggestion: Line 101, please change “insufficient” to “limited” or something similar but less subjective.

Reviewer #3 (Remarks to the Author):

This revised manuscript is improved over the previous version. However, there are still a few points that probably should be addressed by minor revision prior to publication. They follow, indexed by Line number (L).

1. L46-47. Is the phrase "and an increase of warming rates at both intermediate depths of 700–2000 47 m, and in the deeper ocean" supported by the references that follow? Is there a primary reference for warming below 2000 m amongst them?
2. L104-106. The gridded products all use largely the same database, with variations in quality control, instrument bias corrections, and mapping techniques, correlations length and time scales, etc. So how is the ensemble average "more reliable"? Perhaps the most useful quantity to emerge is really the standard deviation for the ensemble.
3. L126-130 and 207-214. What about the addition of water from land (mostly ice melt)? Doesn't that also accelerate over the time period in question? How much of the acceleration in sea level rise is due to acceleration in movement of water from land to the ocean?
4. L279-282. These statements may be true in terms of static water mass definitions based on fixed density (or temperature, or salinity) ranges. However, when a mode water is defined as a low PV region of the water column at a certain location (for instance just north of the SubAntarctic Front for the SubAntarctic Mode Water), or an intermediate water as being centered around a feature (such as a salinity minimum north of the Antarctic Circumpolar Current for Antarctic Intermediate Water), what we see is that the density, temperature, and perhaps salinity of these water masses have shifted. So phenomenologically, the conclusions here seem misleading, and may confuse some readers. They need to be revised.
5. L432-435. Subsurface temperature and salinity changes are not solely governed by ventilation time scales. Isopycnal heave, owing to changes communicated by planetary (e.g., Rossby and Kelvin) waves (e.g., Masada et al., 2010, Science, doi: 10.1126/science.1188703) can also effect these changes.

RESPONSE TO REVIEWER COMMENTS:

Below reviewer comments have been copy-pasted in black font, and our responses are indicated in blue font. Line numbers in our responses refer to the revised manuscript (except where indicated).

Reviewer #1 (Remarks to the Author):

The paper is ready to publish, with only one very minor language editing suggestion: Line 101, please change “insufficient” to “limited” or something similar but less subjective.

Response: We have now changed the word “insufficient” to “limited”, as per the reviewer's suggestion. Thank you. And thanks to Reviewer #1 for their detailed and insightful comments across the previous two rounds of review, which have helped to improve the paper.

Reviewer #3 (Remarks to the Author):

This revised manuscript is improved over the previous version. However, there are still a few points that probably should be addressed by minor revision prior to publication. They follow, indexed by Line number (L).

Response: We thank Reviewer #3 for their detailed and insightful comments in these three rounds of review. We believe these comments have significantly helped us to improve the paper.

1. L46-47. Is the phrase "and an increase of warming rates at both intermediate depths of 700–2000 m, and in the deeper ocean" supported by the references that follow? Is there a primary reference for warming below 2000 m amongst them?

Response: Thanks for pointing this out. Yes, references 5, 7, and 9–11 all support the statement of warming at intermediate depths of the ocean, and references 5, 7, and 10 further support the statement of warming in the deep ocean below 2000 m. We have now separately cited these supporting references for each statement, and

have relocated the previous reference 43 (Bagnell and DeVries 2021) to this sentence, now cited as new reference 11, at lines 46–47 in the manuscript text:

“..., and an increase of warming rates at both intermediate depths of 700–2000 m^{5,7,9–12}, and in the deeper ocean below 2000 m^{5,7,10,11}.”.

2. L104-106. The gridded products all use largely the same database, with variations in quality control, instrument bias corrections, and mapping techniques, correlations length and time scales, etc. So how is the ensemble average "more reliable"? Perhaps the most useful quantity to emerge is really the standard deviation for the ensemble.

Response: Agreed - thanks. We have now changed the word “reliable” to “robust” to make this statement less subjective. A statement regarding the standard deviation of the ensemble follows in the subsequent sentence (lines 106–108):

“The uncertainty of this estimate is taken to be ± 2 standard deviations ($\pm 2\sigma$), to indicate the degree of agreement across the products.”.

3. L126-130 and 207-214. What about the addition of water from land (mostly ice melt)? Doesn't that also accelerate over the time period in question? How much of the acceleration in sea level rise is due to acceleration in movement of water from land to the ocean?

Response: Thanks for pointing this out. The ice melt from glaciers, the Greenland and Antarctic ice sheets, and any land water storage changes has been estimated to account for 19.4%, 15.2%, 8.6%, and 10.9% of global sea-level rise over the period 1993–2018, respectively, while the remaining 45.9% is due to thermal expansion effects of the ocean (Table 9.5 of the latest assessment report of IPCC Working Group I, ref. 1). Both the total sea-level rise and the thermosteric sea-level rise have been increasing across the altimeter measurement era 1993–2018 (compare the rate of sea-level rise over 1993–2018 and 2006–2018 in Table 9.5 of ref. 1, and see Fig. 3 of Cazenave and Moreira (2022, ref. 42)). We have now revised the sentence on lines 126–130 as:

“This may impact the estimated increase in global ocean warming over the past three decades (Fig. 1). However, the rate of global mean sea-level rise has also been increasing since 1993 based on an independent estimate from satellite altimeter data^{1,35}, providing confidence in our results given that half of the global sea surface height increase is due to thermal expansion of the ocean since altimeter measurements began¹.”,

and have included clarifications of the ocean mass component of sea-level rise on lines 207–209:

“An independent measure of OHC changes can be found in altimeter measurements of sea-level if salinity adjustments remain small *and the ocean mass component of sea-level change is known and accounted for.*”,

and on lines 215–218:

“The mass component of sea-level change, due to land ice melt (including glaciers, and the Greenland and Antarctic ice sheets) and land water storage change, has also significantly increased across the altimeter era, but display a more uniform pattern of sea-level rise compared to the thermal expansion component^{41,42}.”.

4. L279-282. These statements may be true in terms of static water mass definitions based on fixed density (or temperature, or salinity) ranges. However, when a mode water is defined as a low PV region of the water column at a certain location (for instance just north of the SubAntarctic Front for the SubAntarctic Mode Water), or an intermediate water as being centered around a feature (such as a salinity minimum north of the Antarctic Circumpolar Current for Antarctic Intermediate Water), what we see is that the density, temperature, and perhaps salinity of these water masses have shifted. So phenomenologically, the conclusions here seem misleading, and may confuse some readers. They need to be revised.

Response: Thanks for the suggestions. To better clarify what we can see from either the density-based definition or the stratification- or salinity-based definition, we have now included a statement of this point on lines 291–296:

“In addition to density-based constraints, alternative criteria like stratification or salinity within specific locations are often used to define mode and intermediate

waters²⁶. These criteria also allow examination of changes in the density and temperature of water masses. Within this framework, the poleward and downward migration of isopycnals over the subtropical oceans and the Southern Ocean can serve as an indication of the warming and lightening of mode and intermediate waters (Supplementary Fig. 2).”.

5. L432-435. Subsurface temperature and salinity changes are not solely governed by ventilation time scales. Isopycnal heave, owing to changes communicated by planetary (e.g., Rossby and Kelvin) waves (e.g., Masada et al., 2010, Science, doi: 10.1126/science.1188703) can also effect these changes.

Response: Thanks for pointing this out. We have now included a citation to Li et al. (2023), Gunn et al. (2023), Purkey and Johnson (2010), and Masuda et al. (2010), and have added a clarification of this point in the manuscript text, lines 442–446:

“Waters deeper than 2000 m are generally not ventilated rapidly enough to have experienced the last 50 years of global warming, but note that both the abyssal overturning slowdown^{72,73} and oceanic internal waves⁷⁴ can result in bottom ocean warming over these time-scales, as seen in observations⁷⁵.”.